# ALMOST SURE CONVERGENCE OF AVERAGE REWARD TEMPORAL DIFFERENCE LEARNING

## ABSTRACT

Tabular average reward Temporal Difference (TD) learning is perhaps the simplest and the most fundamental policy evaluation algorithm in average reward reinforcement learning. After at least 25 years since its discovery, we are finally able to provide a long-awaited almost sure convergence analysis. Namely, we are the first to prove that, under very mild conditions, tabular average reward TD converges almost surely to a sample-path dependent fixed point. Key to this success is a new general stochastic approximation result concerning nonexpansive mappings with Markovian and additive noise, built on recent advances in stochastic Krasnoselskii-Mann iterations.

## 1 INTRODUCTION

Temporal Difference learning (TD, Sutton (1988)) is the most fundamental algorithm in Reinforcement Learning (RL, Sutton & Barto (2018)). In this paper, we investigate the almost sure convergence of TD in its simplest form with a tabular representation, in average reward Markov Decision Processes (MDPs, Bellman (1957); Puterman (2014)). Namely, we investigate the following iterative updates

$$J_{t+1} = J_t + \beta_{t+1}(R_{t+1} - J_t), \qquad \text{(Average Reward TD)}$$
$$v_{t+1}(S_t) = v_t(S_t) + \alpha_{t+1}(R_{t+1} - J_t + v_t(S_{t+1}) - v_t(S_t)),$$

where $\{S_0, R_1, S_1, \dots\}$ is a sequence of states and rewards from an MDP with a fixed policy and a finite state space $\mathcal{S}$, $J_t \in \mathbb{R}$ is the scalar estimate of the average reward, $v_t \in \mathbb{R}^{|\mathcal{S}|}$ is the tabular value estimate, and $\{\alpha_t, \beta_t\}$ are learning rates. This iterative update algorithm, known as average reward TD, dates back to at least Tsitsiklis & Roy (1999). Surprisingly, despite its simplicity and fundamental importance, its almost sure convergence had not been established in the 25 years since its inception until this work. Even more surprisingly, the theoretical analysis of average reward TD with linear function approximation has seen more progress than that of the tabular version we consider here. In this paper, after presenting the necessary background in Section 2, we will elaborate on the difficulty in analyzing tabular average reward TD with existing techniques in Section 3, offering insight into why progress on this topic has been unexpectedly slow. Then we proceed to our central contribution, where we prove that under mild conditions, the iterates $\{v_t\}$ in (Average Reward TD) converge almost surely to a sample-path-dependent fixed point.

This almost sure convergence is achieved by extending recent advances in the convergence analysis of Stochastic Krasnoselskii-Mann (SKM) iterations (Bravo et al., 2019; Bravo & Cominetti, 2024) to settings with Markovian and additive noise. This line of research originates from the seminal work Cominetti et al. (2014), which introduces a novel fox-and-hare race model to analyze Krasnoselskii-Mann (KM) iterations (Krasnosel'skii, 1955). By extending this line of work to Markovian settings, we not only establish the almost sure convergence of average reward TD, but also pave the way for further analysis of other RL algorithms through the lens of SKM iterations.

## 2 BACKGROUND

In this paper, all vectors are column. We use $\|\cdot\|$ to denote a generic operator norm and use $e$ to denote an all-one vector. We use $\|\cdot\|_2$ and $\|\cdot\|_\infty$ to denote $\ell_2$ norm and infinity norm respectively. We use $\mathcal{O}(\cdot)$ to hide deterministic constants for simplifying presentation, while the letter $\zeta$ is reserved for sample-path dependent constants.

In reinforcement learning, we consider an MDP with a finite state space $\mathcal{S}$, a finite action space $\mathcal{A}$, a reward function $r : \mathcal{S} \times \mathcal{A} \to \mathbb{R}$, a transition function $p : \mathcal{S} \times \mathcal{S} \times \mathcal{A} \to [0, 1]$, an initial distribution $p_0 : \mathcal{S} \to [0, 1]$. At time step 0, an initial state $S_0$ is sampled from $p_0$. At time $t$, given the state $S_t$, the agent samples an action $A_t \sim \pi(\cdot|S_t)$, where $\pi : \mathcal{A} \times \mathcal{S} \to [0, 1]$ is the policy being followed by the agent. A reward $R_{t+1} \doteq r(S_t, A_t)$ is then emitted and the agent proceeds to a successor state $S_{t+1} \sim p(\cdot|S_t, A_t)$. In the rest of the paper, we will assume the Markov chain $\{S_t\}$ induced by the policy $\pi$ is irreducible and thus adopts a unique stationary distribution $d_\mu$. The average reward (a.k.a. gain, Puterman (2014)) is defined as

$$\bar{J}_\pi \doteq \lim_{T \to \infty} \tfrac{1}{T} \sum_{t=1}^{T} \mathbb{E}\left[R_t\right].$$

Correspondingly, the differential value function (a.k.a. bias, Puterman (2014)) is defined as

$$v_\pi(s) \doteq \lim_{T \to \infty} \tfrac{1}{T} \sum_{\tau=1}^{T} \mathbb{E}\left[\sum_{i=1}^{\tau} (R_{t+i} - \bar{J}_\pi) \mid S_t = s\right].$$

The corresponding Bellman equation (a.k.a. Poisson's equation) is then

$$v = r_\pi - \bar{J}_\pi e + P_\pi v, \tag{1}$$

where $v \in \mathbb{R}^{|\mathcal{S}|}$ is the free variable, $r_\pi \in \mathbb{R}^{|\mathcal{S}|}$ is the reward vector induced by the policy $\pi$, i.e., $r_\pi(s) \doteq \sum_a \pi(a|s) r(s, a)$, and $P_\pi \in \mathbb{R}^{|\mathcal{S}| \times |\mathcal{S}|}$ is the transition matrix induced by the policy $\pi$, i.e., $P_\pi(s, s') \doteq \pi(a|s) p(s'|s, a)$. It is known (Puterman, 2014) that all solutions to (1) form a set

$$\mathcal{V}_* \doteq \{v_\pi + ce \mid c \in \mathbb{R}\}. \tag{2}$$

The policy evaluation problem in average reward MDPs is to estimate $v_\pi$, perhaps up to a constant offset $ce$. In view of (1) and inspired by the success of TD in the discounted setting (Sutton, 1988), Tsitsiklis & Roy (1999) use (Average Reward TD) to estimate $v_\pi$ (up to a constant offset). In (Average Reward TD), $J_t$ estimates the average reward $\bar{J}_\pi$. Its learning rate, $\beta_t$, does not need to be the same as $\alpha_t$, the learning rate for updating the differential value function estimation.

## 3 HARDNESS OF AVERAGE REWARD TD

To elaborate on the hardness in analyzing (Average Reward TD), we first rewrite it in a compact form. Define the augmented Markov chain $Y_{t+1} \doteq (S_t, A_t, S_{t+1})$. It is easy to see that $\{Y_t\}$ evolves in the finite space $\mathcal{Y} \doteq \{(s, a, s') \mid \pi(a|s) > 0, p(s'|s, a) > 0\}$. We then define a function $H : \mathbb{R}^{|\mathcal{S}|} \times \mathcal{Y} \to \mathbb{R}^{|\mathcal{S}|}$ by defining the $s$-th element of $H(v, (s_0, a_0, s_1))$ as

$$H(v, (s_0, a_0, s_1))[s] \doteq \mathbb{I}\{s = s_0\}\left(r(s_0, a_0) - \bar{J}_\pi + v(s_1) - v(s_0)\right) + v(s).$$

Then, the update to $\{v_t\}$ in (Average Reward TD) can then be expressed as

$$v_{t+1} = v_t + \alpha_{t+1}\left(H(v_t, Y_{t+1}) - v_t + \epsilon_{t+1}\right). \tag{3}$$

Here, $\epsilon_{t+1} \in \mathbb{R}^{|\mathcal{S}|}$ is the random noise vector defined as $\epsilon_{t+1}(s) \doteq \mathbb{I}\{s = S_t\}(J_t - \bar{J}_\pi)$. This $\epsilon_{t+1}$ is the current estimate error of the average reward estimator $J_t$. Intuitively, the indicator $\mathbb{I}\{s = S_t\}$ reflects the asynchronous nature of (Average Reward TD). For each $t$, only the $S_t$-indexed element in $v_t$ is updated. To better analyze (3), we investigate the expectation of $H$. We define

$$h(v) \doteq \mathbb{E}_{s_0 \sim d_\mu, a_0 \sim \pi(\cdot|s_0), s_1 \sim p(\cdot|s_0, s_1)}\left[H(v, (s_0, a_0, s_1))\right] \tag{4}$$
$$= D(r_\pi - \bar{J}_\pi e + P_\pi v - v) + v,$$

where $D \in \mathbb{R}^{|\mathcal{S}| \times |\mathcal{S}|}$ is a diagonal matrix with the diagonal being the stationary distribution $d_\mu$. Then we can write the limiting ODE of (3) as

$$\frac{\mathrm{d}v(t)}{\mathrm{d}t} = h(v(t)) - v(t). \tag{5}$$

**Hardness in Stability.** Stability (i.e., $\sup_t \|v_t\| < \infty$ almost surely) is a necessary condition for almost sure convergence. In ODE based stochastic approximation methods to establish almost sure convergence, the first step is usually to establish the stability (Benveniste et al., 1990; Kushner & Yin,

2003; Borkar, 2009). The ODE@$\infty$ technique (Borkar & Meyn, 2000; Borkar et al., 2021; Liu et al., 2024) is perhaps one of the most powerful stability techniques in RL, which considers the function

$$h_\infty(v) \doteq \lim_{c \to \infty} \frac{h(cv)}{c} = D(P_\pi - I)v + v.$$

Correspondingly, the ODE@$\infty$ is defined as

$$\frac{\mathrm{d}v(t)}{\mathrm{d}t} = h_\infty(v(t)) - v(t) = D(P_\pi - I)v(t). \tag{6}$$

If the ODE (6) is globally asymptotically stable, existing results such as Borkar et al. (2021); Liu et al. (2024) can be used to establish the desired stability of $\{v_t\}$. Unfortunately, the vector $ce$ with any $c \in \mathbb{R}$ is an equilibrium of (6), so it cannot be globally asymptotically stable. This problem comes from the lack of a discounting factor in average reward MDPs. In the discounted setting, with a discount factor $\gamma \in [0, 1)$, the corresponding ODE@$\infty$ is $\frac{\mathrm{d}v(t)}{\mathrm{d}t} = D(\gamma P_\pi - I)v(t)$. It is well known that $D(\gamma P_\pi - I)$ is negative definite (Tsitsiklis & Roy, 1996) and therefore Hurwitz. As a result, it is globally asymptotically stable. Alternatively, if a discount factor is present, an inductive argument can also be used to establish stability following the method in Gosavi (2006). However, in the average reward setting, there is no discounting, so neither the Hurwitz argument nor the inductive argument applies.

**Hardness in Convergence.** Suppose we were somehow able to establish the desired stability, then standard stochastic approximation results can be used to show that $\{v_t\}$ converge almost surely to a bounded invariant set of the ODE (5), or more precisely speaking, a possibly sample-path dependent compact connected internally chain transitive invariant set[1] (Kushner & Yin, 2003; Borkar, 2009). Unfortunately, we are not aware of any finer characterization of this set. Even if it was proved that this set must be a subset of $\mathcal{V}_*$ in (2) (we are not aware of any such proof yet), the best we could say is still that $\{v_t\}$ converges to this set. It is still possible that $\{v_t\}$ oscillates within this set or around the neighborhood of this set and never settles down on any particular fixed point. This gives rise to the central open question that this paper aims to answer:

can we prove that $\{v_t\}$ converge almost surely to a single fixed point in $\mathcal{V}_*$?

We shall give an affirmative answer shortly. We note that this affirmative answer is quite intuitive. Notice that

$$h(v) = (I + D(P_\pi - I))v + D(r_\pi - \bar{J}_\pi e). \tag{7}$$

It is easy to verify that $I + D(P_\pi - I)$ is a stochastic matrix. It then follows that $\|I + D(P_\pi - I)\|_\infty = 1$. As a result, the operator $h$ is a nonexpansive mapping w.r.t. $\|\cdot\|_\infty$ (Lemma 3). Since the ODE (5) can be expressed as

$$\frac{\mathrm{d}v(t)}{\mathrm{d}t} = h(v(t)) - v(t),$$

the nonexpansivity of $h$ confirms that any solution $v(t)$ to the ODE (5) will converge to an initial value dependent fixed point in $\mathcal{V}_*$ (Theorem 3.1 of Borkar & Soumyanatha (1997)). So intuitively, if $\{v_t\}$ approximates a solution $v(t)$ well, it should also converge to a single fixed point. However, existing ODE-based convergence analysis is limited, as it can only establish convergence to a bounded invariant set. This difficulty stems from two sources. The first is still the lack of the discount factor $\gamma$ in average reward MDPs. Otherwise, $h$ can easily be a contraction, and the ODE (5) would then be globally asymptotically stable. As a result, the invariant set would be a singleton. The second is the lack of a reference value (Abounadi et al., 2001). In a recent differential TD algorithm (Wan et al., 2021b), the corresponding ODE is

$$\frac{\mathrm{d}v(t)}{\mathrm{d}t} = r_\pi - ee^\top v(t) + P_\pi v(t) - v(t), \tag{8}$$

where $ee^\top v(t)$ serves as a reference value to stabilize the trajectories. Wan et al. (2021b) prove that the ODE (8) is globally asymptotically stable. As a result, its invariant set is a singleton. We note

---

[1]We refer the reader to Chapter 2 of Borkar (2009) for the definition of a connected internally chain transitive invariant set.

that to use this reference value technique in learning algorithms such as Abounadi et al. (2001); Wan et al. (2021b), we have to replace the learning rate $\alpha_t$ with a count-based learning rate $\alpha_{n(Y_t, t)}$. Here $n(y, t) \doteq \sum_{\tau=0}^{t} \mathbb{I}\{Y_\tau = y\}$ counts the number of visits to the a state $y$ until time $t$. The detailed benefits (in terms of convergence) of this count-based learning rate are beyond the scope of this work, and we refer the reader to Chapter 7 of Borkar (2009) for more discussion. Here, we only argue that this count based learning rate is unnatural compared with the straightforward $\alpha_t$, and it cannot be used once function approximation is introduced. Alternatively, to make use of this reference value technique without a count-based learning rate, one has to resort to synchronous updates, where at each time step $t$, all elements of $v_t$, not just $v(S_t)$, are updated (Zhang et al., 2021; Bravo & Cominetti, 2024). Such synchronous updates are impossible when we have access to only one Markovian data stream $\{S_0, A_0, R_0, S_1, \dots\}$.

**Hardness with Linear Function Approximation.** The (Average Reward TD) has also been extended to linear function approximation (Tsitsiklis & Roy, 1999; Konda & Tsitsiklis, 1999; Wu et al., 2020; Zhang et al., 2021). Unfortunately, the results in linear function approximation do not contribute much to the understanding of the tabular version. In the following paragraphs, we elaborate on this surprising fact.

Instead of using a look-up table $v \in \mathbb{R}^{|\mathcal{S}|}$ to store the value estimate, the idea of linear function approximation is to approximate $v(s)$ with $\phi(s)^\top w$, where $\phi : \mathcal{S} \to \mathbb{R}^K$ is the feature function mapping a state $s$ to a $K$-dimensional feature $\phi(s) \in \mathbb{R}^K$ and $w$ is the learnable weights. Let $\Phi \in \mathbb{R}^{|\mathcal{S}| \times K}$ be the feature matrix, whose $s$-th row is the $\phi(s)^\top$. Then, linear function approximation essentially uses $\Phi w$ to approximate $v$. It is obvious that if $\Phi = I$ (i.e., a one-hot feature encoding is used), linear function approximation degenerates to the tabular method. Thus, one would expect the results in linear function approximation to subsume tabular results. This is true in most settings, but for (Average Reward TD) there is some subtlety. The linear average reward TD (Tsitsiklis & Roy, 1999) updates $\{w_t\}$ iteratively as

$$w_{t+1} = w_t + \alpha_{t+1} \left( R_{t+1} - J_t + \phi(S_{t+1})^\top w_t - \phi(S_t)^\top w_t \right) \phi(S_t),$$

where the update of $\{J_t\}$ is identical to (Average Reward TD). The limiting ODE of this update is,

$$\frac{\mathrm{d}w(t)}{\mathrm{d}t} = \Phi^\top D(P_\pi - I)\Phi v(t) + \Phi^\top D(r_\pi - \bar{J}_\pi e). \tag{9}$$

Unfortunately, the matrix $\Phi^\top D(P_\pi - I)\Phi$ is not necessarily Hurwitz. Consequently, the ODE (9) is not necessarily globally asymptotically stable. The problem still arises from the lack of a discount factor – it is well known that $\Phi^\top D(\gamma P_\pi - I)\Phi$ is negative definite and thus Hurwitz.

Nevertheless, to proceed with the theoretical analysis, besides the standard assumption that $\Phi$ has linearly independent columns, Tsitsiklis & Roy (1999); Konda & Tsitsiklis (1999) further assume that for any $c \in \mathbb{R}, w \in \mathbb{R}^d$, it holds that $\Phi w \neq ce$. Under this assumption, Tsitsiklis & Roy (1999) prove that $\Phi^\top D(P_\pi - I)\Phi$ is negative definite (Wu et al. (2020) assume this negative definiteness directly) and the iterates $\{w_t\}$ converges almost surely. Unfortunately, this additional assumption does not hold in the tabular setting where $\Phi = I$ (apparently, $Ie = e$). As a result, the almost sure convergence in Tsitsiklis & Roy (1999) does not shed light on the behavior of tabular average reward TD. A more recent work Zhang et al. (2021) proves that

$$\min_{\|w\|_2 = 1, w \in E} w^\top \Phi^\top D(P_\pi - I)\Phi w > 0 \tag{10}$$

without requiring $\Phi w \neq ce$, where $E$ is a subspace of $\mathbb{R}^K$. Based on this, Zhang et al. (2021) prove that

$$\mathbb{E}\left[ \left| J_t - \bar{J}_\pi \right|^2 + \|\Pi_E(w_t - w_*)\|_2^2 \right]$$

converges to 0, where $w_*$ is one desired fixed point and $\Pi_E$ denotes the orthogonal projection onto the subspace $E$. Zhang et al. (2021) further provide a convergence rate. This is a significant improvement over Tsitsiklis & Roy (1999), but still not satisfactory in two aspects. First, this result is convergence in $L^2$, not almost sure convergence. It is well known that almost sure convergence and $L^2$ convergence usually do not imply each other. It is also not clear whether (10) can be used to establish an almost sure convergence under the presence of the projection. Second, if we consider the

tabular case where $\Phi = I$, then according to the Appendix A.1 of Zhang et al. (2021), $E$ becomes the orthogonal complement of $\{ce \mid c \in \mathbb{R}\}$. Even if we were able to similarly prove that almost surely $\lim_{t\to\infty} \|\Pi_E(v_t - v_*)\|_2 = 0$ for some $v_*$ (again, it is not clear how this can be done), we could still only conclude that $\{v_t\}$ converges to a set, not a point.

To summarize, recent advances with linear average reward TD present insightful results, but those results do not say much (if anything) about the almost sure convergence of tabular average reward TD.

**Hardness in Stochastic Krasnoselskii-Mann Iterations.** Having elaborated on the hardness in analyzing (Average Reward TD) with ODE-based approaches, we now resort to an alternative approach, the Stochastic Krasnoselskii-Mann (SKM) iterations. In its simplest and deterministic form, Krasnoselskii-Mann (KM) iterations study the convergence of iterates

$$x_{t+1} = x_t + \alpha_{t+1}(Tx_t - x_t), \tag{KM}$$

where $T$ is some nonexpansive mapping. Since we have already demonstrated that $h$ is nonexpansive in $\|\cdot\|_\infty$, SKM appears promising in analyzing (Average Reward TD). It, however, turns out that the current state of results for SKM iterations is insufficient for proving the almost sure convergence of (Average Reward TD). We elaborate on this fact here.

Earlier works on the convergence of (KM) typically require that the operator $T : C \to C$ has a compact image, i.e., $T(C)$ is a compact subset of $C$. Under some other restrictive conditions, Krasnosel'skii (1955) first proves the convergence of (KM) to a fixed point of $T$. This result is further generalized by Edelstein (1966); Schaefer (1957); Ishikawa (1976); Reich (1979). More recently, Cominetti et al. (2014) use a novel fox-and-hare model to connect KM iterations with Bernoulli random variables, providing a sharper convergence rate for $\|x_k - Tx_k\| \to 0$.

However, in many scenarios such as RL, requiring an algorithm to satisfy the exact form of (KM) is usually not plausible. Instead, some noise may appear. This gives rise to the study of the inexact KM iterations (IKM).

$$x_{t+1} = x_t + \alpha_{t+1}(Tx_t - x_t + e_{t+1}), \tag{IKM}$$

where $\{e_t\}$ is a sequence of deterministic noise. Bravo et al. (2019) extend Cominetti et al. (2014) and establish the convergence of (IKM), under some mild conditions on $\{e_t\}$.

However, a deterministic noise is still not desirable in many problems. To this end, a stochastic version of (IKM) is studied, which considers the iterates

$$x_{t+1} = x_t + \alpha_{t+1}(Tx_t - x_t + M_{t+1}), \tag{SKM}$$

where $\{M_t\}$ is a Martingale difference sequence. Under mild conditions, Bravo & Cominetti (2024) prove the almost sure convergence of (SKM) to a fixed point of $T$.

Using (3) and (4), we can write (Average Reward TD) as,

$$v_{t+1} = v_t + \alpha_{t+1}(h(v_t) - v_t + H(v_t, Y_{t+1}) - h(v_t) + \epsilon_{t+1}), \tag{11}$$

where we recall that $h$ is nonexpansive in $\|\cdot\|_\infty$. This, however, does not fit into (SKM). First, there is an additive stochastic noise $\{\epsilon_t\}$. Second, the sequence $\{H(v_t, Y_{t+1}) - h(v_t)\}$ is not a Martingale difference sequence. If the sequence of noise $\{Y_t\}$ was i.i.d., then $\{H(v_t, Y_{t+1}) - h(v_t)\}$ would have been a Martingale difference sequence. But unfortunately, in (Average Reward TD), the sequence $\{Y_t\}$ is a Markov chain, far from being i.i.d. Moreover, the noise $\{\epsilon_t\}$ is now stochastic. So (IKM) concerning a deterministic noise would not apply either. These demonstrate the hardness in analyzing (Average Reward TD) with existing (SKM) results.

Nevertheless, this motivates us to extend the results from Bravo & Cominetti (2024) to study (SKM) with Markovian and additive noise, in the form of (11).

# 4 STOCHASTIC KRASNOSELSKII-MANN ITERATIONS WITH MARKOVIAN AND ADDITIVE NOISE

As promised, we are now ready to extend the analysis of (SKM) in Bravo et al. (2019); Bravo & Cominetti (2024) to SKM with Markovian and additive noise. Namely, we consider the following

iterates

$$x_{n+1} = x_n + \alpha_{n+1}\left(H(x_n, Y_{n+1}) - x_n + \epsilon_{n+1}^{(1)}\right). \quad \text{(SKM with Markovian and Additive Noise)}$$

Here $\{x_n\}$ are stochastic vectors evolving in $\mathbb{R}^d$, $\{Y_n\}$ is a Markov chain evolving in a finite state space $\mathcal{Y}$, $H : \mathbb{R}^d \times \mathcal{Y} \to \mathbb{R}^d$ defines the update, $\left\{\epsilon_{n+1}^{(1)}\right\}$ is a sequence of stochastic noise evolving in $\mathbb{R}^d$, and $\{\alpha_n\}$ is a sequence of deterministic learning rates. We make the following assumptions.

**Assumption 4.1** (Ergodicity). *The Markov chain $\{Y_n\}$ is irreducible and aperiodic.*

The Markov chain $\{Y_n\}$ thus adopts a unique invariant distribution, denoted as $d_\mu$. We use $P$ to denote the transition matrix of $\{Y_n\}$.

**Assumption 4.2** (1-Lipschitz). *The function $H$ is 1-Lipschitz continuous in its first argument w.r.t. some operator norm $\|\cdot\|$ and uniformly in its second argument, i.e., for any $x, x', y$, it holds that*

$$\|H(x, y) - H(x', y)\| \le \|x - x'\|.$$

This assumption has two important implication. First, it implies that $H(x, y)$ can grow at most linearly. Indeed, let $x' = 0$, we get $\|H(x, y)\| \le \|H(0, y)\| + \|x\|$. Define $C_H \doteq \max_y \|H(0, y)\|$, we get

$$\|H(x, y)\| \le C_H + \|x\|. \tag{12}$$

Second, define the function $h : \mathbb{R}^d \to \mathbb{R}^d$ as the expectation of $H$ over the stationary distribution $d_\mu$:

$$h(x) \doteq \mathbb{E}_{y \sim d_\mu}[H(x, y)].$$

We then have that $h$ is nonexpansive. Namely,

$$\|h(x) - h(x')\| \le \sum_y d_\mu(y)\|H(x, y) - H(x', y)\| \le \|x - x'\|. \tag{13}$$

This $h$ is exactly the nonexpansive operator in the SKM literature. We of course need to assume that the problem is solvable.

**Assumption 4.3** (Fixed Points). *The nonexpansive operator $h$ adopts at least one fixed point.*

We use $\mathcal{X}_* \ne \emptyset$ to denote the set of the fixed points of $h$.

**Assumption 4.4** (Learning Rate). *The learning rate $\{\alpha_n\}$ has the form*

$$\alpha_n = \tfrac{1}{(n+1)^b}, \alpha_0 = 0,$$

*where $b \in (\frac{4}{5}, 1]$.*

The primary motivation for requiring $b \in (\frac{4}{5}, 1]$ is that our learning rates $\alpha_n$ need to decrease quickly enough for certain key terms in the proof to be finite. The specific need for $b > \frac{4}{5}$ can be seen in the proof of (35) in Lemma 8. We now impose assumptions on the additive noise.

**Assumption 4.5** (Additive Noise).

$$\sum_{k=1}^{\infty} \alpha_k \left\|\epsilon_k^{(1)}\right\| < \infty \quad a.s., \tag{14}$$

$$\mathbb{E}\left[\left\|\epsilon_n^{(1)}\right\|^2\right] = \mathcal{O}\left(\tfrac{1}{n}\right). \tag{15}$$

The first part of Assumption 4.5 can be interpreted as a requirement that the total amount of additive noise remains finite, akin to the assumption on $e_t$ in (IKM) in Bravo et al. (2019). Additionally, we impose a condition on the second moment of this noise, requiring it to converge at the rate $\mathcal{O}\left(\tfrac{1}{n}\right)$. While these assumptions on $\epsilon_n^{(1)}$ may seem restrictive, we introduce $\epsilon_n^{(1)}$ because it is essential for proving the convergence of (Average Reward TD). It is worth noting that even if $\epsilon_n^{(1)}$ were absent, our work would still extend the results of (Bravo & Cominetti, 2024) to cases involving Markovian noise, as the Markovian noise component is already incorporated within $Y_n$, which represents a significant result. We are now ready to present the main result.

**Theorem 1.** *Let Assumptions 4.1 - 4.5 hold. Then the iterates $\{x_n\}$ generated by* (SKM with Markovian and Additive Noise) *satisfy*

$$\lim_{n\to\infty} x_n = x_* \quad a.s.,$$

*where $x_* \in \mathcal{X}_*$ is a possibly sample-path dependent fixed point.*

**Proof** We start with a decomposition of the error $H(x, Y_{n+1}) - h(x)$ using Poisson's equation akin to Métivier & Priouret (1987); Benveniste et al. (1990). Namely, thanks to the finiteness of $\mathcal{Y}$, it is well known (see, e.g., Theorem 17.4.2 of Meyn & Tweedie (2012) or Theorem 8.2.6 of Puterman (2014)) that there exists a function $\nu(x, y) : \mathbb{R}^d \times \mathcal{Y} \to \mathbb{R}^d$ such that

$$H(x, y) - h(x) = \nu(x, y) - (P\nu)(x, y). \tag{16}$$

Here, we use $P\nu$ to denote the function $(x, y) \mapsto \sum_{y'} P(y, y')\nu(x, y')$. The error can then be decomposed as

$$H(x, Y_{n+1}) - h(x) = M_{n+1} + \epsilon_{n+1}^{(2)} + \epsilon_{n+1}^{(3)}, \tag{17}$$

where

$$M_{n+1} \doteq \nu(x_n, Y_{n+2}) - (P\nu)(x_n, Y_{n+1}), \tag{18}$$

$$\epsilon_{n+1}^{(2)} \doteq \nu(x_n, Y_{n+1}) - \nu(x_{n+1}, Y_{n+2}), \tag{19}$$

$$\epsilon_{n+1}^{(3)} \doteq \nu(x_{n+1}, Y_{n+2}) - \nu(x_n, Y_{n+2}). \tag{20}$$

Here $\{M_{n+1}\}$ is a Martingale difference sequence. We then use

$$\xi_{n+1} \doteq \epsilon_{n+1}^{(1)} + \epsilon_{n+1}^{(2)} + \epsilon_{n+1}^{(3)}, \tag{21}$$

to denote all the non-Martingale noise, yielding

$$x_{n+1} = (1 - \alpha_{n+1})x_n + \alpha_{n+1}(h(x_n) + M_{n+1} + \xi_{n+1}).$$

We now define an auxiliary sequence $\{U_n\}$ to capture how the noise evolves

$$U_0 \doteq 0,$$
$$U_{n+1} \doteq (1 - \alpha_{n+1})U_n + \alpha_{n+1}(M_{n+1} + \xi_{n+1}). \tag{22}$$

If we are able to prove that the total noise is well controlled in the following sense

$$\sum_{k=1}^{\infty} \alpha_k \|U_{k-1}\| < \infty \quad a.s., \tag{23}$$

$$\lim_{n\to\infty} \|U_n\| = 0 \quad a.s., \tag{24}$$

then a result from Bravo & Cominetti (2024) concerning the convergence of (IKM) can be applied on each sample path to complete the almost sure convergence proof. The rest of the proof is dedicated to the verification of those two conditions. To this end, we first define shorthand

$$\alpha_{k,n} \doteq \alpha_k \prod_{j=k+1}^{n} (1 - \alpha_j), \; \alpha_{n,n} \doteq \alpha_n. \tag{25}$$

Telescoping (22) then yields

$$U_n = \underbrace{\sum_{k=1}^{n} \alpha_{k,n} M_k}_{\overline{M}_n} + \underbrace{\sum_{k=1}^{n} \alpha_{k,n} \epsilon_k^{(1)}}_{\overline{\epsilon}_n^{(1)}} + \underbrace{\sum_{k=1}^{n} \alpha_{k,n} \epsilon_k^{(2)}}_{\overline{\epsilon}_n^{(2)}} + \underbrace{\sum_{k=1}^{n} \alpha_{k,n} \epsilon_k^{(3)}}_{\overline{\epsilon}_n^{(3)}}. \tag{26}$$

Then, we can upper-bound (23) as

$$\sum_{k=1}^{n} \alpha_k \|U_{k-1}\| \leq \underbrace{\sum_{k=1}^{n} \alpha_k \|\overline{M}_{k-1}\|}_{\overline{\overline{M}}_n} + \underbrace{\sum_{k=1}^{n} \alpha_i \left\|\overline{\epsilon}_{k-1}^{(1)}\right\|}_{\overline{\overline{\epsilon}}_n^{(1)}} + \underbrace{\sum_{k=1}^{n} \alpha_i \left\|\overline{\epsilon}_{k-1}^{(2)}\right\|}_{\overline{\overline{\epsilon}}_n^{(2)}} + \underbrace{\sum_{k=1}^{n} \alpha_k \left\|\overline{\epsilon}_{k-1}^{(3)}\right\|}_{\overline{\overline{\epsilon}}_n^{(3)}}. \tag{27}$$

Lemmas 15, 16, 17, and 18 respectively prove that all terms in (27) are bounded almost surely, which verifies (23).

We now verify (24). This time, rewrite $U_n$ as

$$U_n = -\sum_{k=1}^{n} \alpha_k U_{k-1} + \alpha_k \left( M_k + \epsilon_k^{(1)} + \epsilon_k^{(2)} + \epsilon_k^{(3)} \right).$$

Lemma 19, Assumption 4.5, and Lemmas 20, 21 prove that $\sup_n \left\| \sum_{k=1}^{n} \alpha_k M_k \right\| < \infty$ and $\sup_n \left\| \sum_{k=1}^{n} \alpha_k \epsilon_k^{(j)} \right\| < \infty$ for $j \in \{1, 2, 3\}$ respectively. Together with (26), this means that $\sup_n \|U_n\| < \infty$. In other words, we have established the stability of (22). Then, it can be shown (Lemma 22), using an extension of Theorem 2.1 of Borkar (2009) (Lemma 25), that $\{U_n\}$ converges to the globally asymptotically stable equilibrium of the ODE $\frac{\mathrm{d}U(t)}{\mathrm{d}t} = -U(t)$, which is 0. This verifies (24). Lemma 23 then invokes a result from Bravo & Cominetti (2024) and completes the proof. ∎

## 5 AVERAGE REWARD TEMPORAL DIFFERENCE LEARNING

We are now ready to prove the convergence of (Average Reward TD). Throughout the rest of the section, we utilize the following assumption.

**Assumption 5.1** (Ergodicity). *Both $\mathcal{S}$ and $\mathcal{A}$ are finite. The Markov chain $\{S_t\}$ induced by the policy $\pi$ is aperiodic and irreducible.*

**Theorem 2.** *Let Assumption 5.1 hold. Consider the learning rates in the form of $\alpha_t = \frac{1}{(t+1)^b}, \beta_t = \frac{1}{t}$ with $b \in (\frac{4}{5}, 1]$. Then the iterates $\{v_t\}$ generated by (Average Reward TD) satisfy*

$$\lim_{t \to \infty} v_t = v_* \quad a.s.,$$

*where $v_* \in \mathcal{V}_*$ is a possibly sample-path dependent fixed point.*

**Proof** We proceed via verifying assumptions of Theorem 1. In particular, we consider the compact form (3). Under Assumption 5.1, it is obvious that $\{Y_t\}$ is irreducible and aperiodic and adopts a unique stationary distribution.

To verify Assumption 4.2, we demonstrate that $H$ is $1-$Lipschitz in $v$ w.r.t $\|\cdot\|_\infty$. For notation simplicity, let $y = (s_0, a_0, s_1)$. We have,

$$H(v, y)[s] - H(v', y)[s] = \mathbb{I}\{s = s_0\}(v(s_1) - v'(s_1) - v(s_0) + v'(s_0)) + v(s) - v'(s).$$

Separating cases based on $s$, if $s \neq s_0$, we have

$$|H(v, y)[s] - H(v', y)[s]| = |v(s) - v'(s)| \leq \|v - v'\|_\infty.$$

For the case when $s = s_0$, we have

$$|H(v, y)[s] - H(v', y)[s]| = |v(s_1) - v'(s_1)| \leq \|v - v'\|_\infty.$$

Therefore

$$\|H(v, y) - H(v, y)\|_\infty = \max_{s \in \mathcal{S}} |H(v, y)[s] - H(v', y)[s]| \leq \|v - v'\|_\infty.$$

It is well known that the set of solutions to Poisson's equation $\mathcal{V}_*$ defined in (2) is non-empty (Puterman, 2014), verifying Assumption 4.3. Assumption 4.4 is directly met by the definition of $\alpha_t$.

To verify Assumption 4.5, we first notice that for (Average Reward TD), we have $\left\| \epsilon_t^{(1)} \right\|_\infty = \left| \bar{J}_\pi - J_t \right|$. It is well-known from the ergodic theorem that $J_t$ converges to $\bar{J}_\pi$ almost surely. To verify Assumption 4.5, however, requires both an almost sure convergence rate and an $L^2$ convergence rate. To this end, we rewrite the update of $\{J_t\}$ as

$$J_{t+1} = J_t + \beta_{t+1} \left( R_{t+1} + \gamma J_t \phi(S_{t+1}) - J_t \phi(S_t) \right) \phi(S_t),$$

where we define $\gamma \doteq 0$ and $\phi(s) \doteq 1 \, \forall s$. It is now clear that the update of $\{J_t\}$ is a special case of linear TD in the discounted setting (Sutton, 1988). Given our choice of $\beta_t = \frac{1}{t}$, the general result about the almost sure convergence rate of linear TD (Theorem 1 of Tadić (2002)) ensures that

$$\left| J_t - \bar{J}_\pi \right| \leq \frac{\zeta_2 \sqrt{\ln \ln t}}{\sqrt{t}} \quad \text{a.s.,}$$

where $\zeta_2$ is a sample-path dependent constant. This immediately verifies (14). We do note that this almost sure convergence rate can also be obtained via a law of the iterated logarithm for Markov chains (Theorem 17.0.1 of Meyn & Tweedie (2012)). The general result about the $L^2$ convergence rate of linear TD (Theorem 11 of Srikant & Ying (2019)) ensures that

$$\mathbb{E}\left[ \left| J_t - \bar{J}_\pi \right|^2 \right] = \mathcal{O}\left( \frac{1}{t} \right).$$

This immediately verifies (15) and completes the proof. ∎

## 6 RELATED WORK

It is now clear that our success fundamentally originates from the novel fox-and-hare racing model introduced by Cominetti et al. (2014). This fox-and-hare model is too complicated to be detailed here, but it is for sure an entirely different paradigm from the ODE and Lyapunov based methods in RL (Bertsekas & Tsitsiklis, 1996; Konda & Tsitsiklis, 1999; Borkar & Meyn, 2000; Srikant & Ying, 2019; Borkar et al., 2021; Chen et al., 2021; Zhang et al., 2022; Meyn, 2022; Zhang et al., 2023; Liu et al., 2024; Meyn, 2024). Bravo & Cominetti (2024) is the first to introduce this SKM based method in RL, which analyzes a synchronous version of RVI $Q$-learning (Abounadi et al., 2001). The method in Bravo & Cominetti (2024) is only applicable to synchronous RL algorithms because it requires Martingale difference noise. By extending Bravo & Cominetti (2024) to Markovian noise, we are the first to use the SKM method to analyze asynchronous RL algorithms.

Poisson's equation has been very powerful in dealing with Markovian noise. In particular, the noise representation (17) is not new. However, our work bounds the error terms in (17) differently from previous works concerning the almost sure convergence. Namely, Benveniste et al. (1990); Konda & Tsitsiklis (1999) use stopping times to bound the error terms while Borkar et al. (2021) use scaled iterates. Instead, we rely on the 1-Lipschitz continuity (Assumption 4.2) to bound the growth of the error terms directly. Moreover, previous works with such error decomposition (e.g., Benveniste et al. (1990); Konda & Tsitsiklis (1999); Borkar et al. (2021)) usually only need to bound terms like $\sum_k \alpha_k \epsilon_k^{(1)}$. For our setup, besides $\sum_k \alpha_k \epsilon_k^{(1)}$, we also need to bound terms like $\bar{\epsilon}_n^{(1)} = \sum_k \alpha_{k,n} \epsilon_k^{(1)}$ and $\bar{\bar{\epsilon}}_n^{(1)} = \sum_i \alpha_i \left\| \bar{\epsilon}_{k-1}^{(1)} \right\|$, which appear novel and more challenging.

The (Average Reward TD) algorithm has inspired the design of many other temporal difference algorithms for average reward MDPs, for both policy evaluation and control, including Konda & Tsitsiklis (1999); Yang et al. (2016); Wan et al. (2021a); Zhang & Ross (2021); Wan et al. (2021b); He et al. (2022); Saxena et al. (2023). We envision that our work will shed light on the almost sure convergence of those follow-up algorithms.

## 7 CONCLUSION

After more than 25 years since the discovery of (Average Reward TD), we have finally established its almost sure convergence to a potentially sample-path dependent fixed point. This result highlights the underappreciated strength of SKM iterations, a tool whose potential is often overlooked in the RL community. Addressing several follow-up questions could open the door to proving the convergence of many other RL algorithms. Do SKM iterations converge in $L^p$? Do they follow a central limit theorem or a law of the iterated logarithm? Can they be extended to two-timescale settings? And can we develop a finite sample analysis for them? Resolving these questions could pave the way for significant advancements across reinforcement learning theory. We leave them for future investigation.

ACKNOWLEDGMENTS

This work is supported in part by the US National Science Foundation (NSF) under grants III-2128019 and SLES-2331904. EB acknowledges support from the NSF Graduate Research Fellowship (NSF-GRFP) under award 1842490.

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

## A   MATHEMATICAL BACKGROUND

**Lemma 3** (Non-expansivity of $h$). *With $h$ defined in* (7)*, we have*

$$\|h(v) - h(v')\|_\infty \le \|v - v'\|_\infty.$$

**Proof** Let

$$A_\pi \doteq (I + D(P_\pi - I)).$$

From (7), we have

$$h(v) - h(v') = A_\pi(v - v').$$

The matrix $A_\pi$ is a row-stochastic matrix. To see that the entries of $A_\pi$ are non-negative, for any diagonal entry, we have

$$A_\pi(i, i) = 1 - d_\mu(i) + d_\mu(i)P_\pi(i, i) \ge 0,$$

and for any off-diagonal entry, we have

$$A_\pi(i, j) = (DP_\pi)(i, j) \ge 0.$$

To see that the row sum of $A_\pi$ is always one, we have

$$\begin{aligned} A_\pi e &= (I - D(I - P_\pi))e, \\ &= Ie - DIe + DP_\pi e, \\ &= e. \end{aligned}$$

Since we have proven $A_\pi$ is a stochastic matrix, we know that $\|A_\pi\|_\infty \le 1$. Therefore,

$$\begin{aligned} \|h(v) - h(v')\|_\infty &= \|A_\pi(v - v')\|_\infty, \\ &\le \|v - v'\|_\infty. \end{aligned}$$

∎

**Lemma 4** (Theorem 2.1 from Bravo & Cominetti (2024)). *Let $\{z_n\}$ be a sequence generated by* (IKM)*. Let Fix$(T)$ denote the set of fixed points of $T$ (assumed to be nonempty). Additionally, let $\tau_n$ be defined according to* (29) *and the real function $\sigma : (0, \infty) \to (0, \infty)$ as*

$$\sigma(y) = \min\{1, 1/\sqrt{\pi y}\}.$$

*If $\kappa \ge 0$ is such that $\|Tz_n - x_0\| \le \kappa$ for all $n \ge 1$, then*

$$\|z_n - Tz_n\| \le \kappa\sigma(\tau_n) + \sum_{k=1}^n 2\alpha_k\|e_k\|\sigma(\tau_n - \tau_k) + 2\|e_{n+1}\|. \tag{28}$$

*Moreover, if $\tau_n \to \infty$ and $\|e_n\| \to 0$ with $S \doteq \sum_{n=1}^\infty \alpha_n\|e_n\| < \infty$, then* (28) *holds with $\kappa = 2\inf_{x \in Fix(T)}\|x_0 - x\| + S$, and we have $\|z_n - Tz_n\| \to 0$ as well as $z_n \to x_*$ for some fixed point $x_* \in Fix(T)$*

**Lemma 5** (Monotonicity of $\alpha_{k,n}$ from Lemma B.1 in Bravo & Cominetti (2024)). *For $\alpha_n = \frac{1}{(n+1)^b}$ with $0 < b \le 1$ and $\alpha_{i,n}$ in* (25)*, we have $\alpha_{k,n} \le \alpha_{k+1,n}$ for $k \ge 1$ so that $\alpha_{k+1,n} \le \alpha_{n,n} = \alpha_n$.*

**Lemma 6** (Lemma B.2 from Bravo & Cominetti (2024)). *For $\alpha_n = \frac{1}{(n+1)^b}$ with $0 < b \le 1$ and $\alpha_{i,n}$ in* (25)*, we have $\sum_{k=1}^n \alpha_{k,n}^2 \le \alpha_{n+1}$ for all $n \ge 1$.*

**Lemma 7** (Monotone Convergence Theorem from Folland (1999)). *Given a measure space $(X, M, \mu)$, define $L^+$ as the space of all measurable functions from $X$ to $[0, \infty]$. Then, if $\{f_n\}$ is a sequence in $L^+$ such that $f_j \le f_{j+1}$ for all $j$, and $f = \lim_{n\to\infty} f_n$, then $\int f d\mu = \lim_{n\to\infty} \int f_n d\mu$.*

## B    ADDITIONAL LEMMAS FROM SECTION 4

In this section, we present and prove the lemmas referenced in Section 4 as part of the proof of Theorem 1. Additionally, we establish several auxiliary lemmas necessary for these proofs.

Additionally, using the learning rates defined in 4.4, we define

$$\tau_n \doteq \sum_{k=1}^{n} \alpha_k (1 - \alpha_k). \tag{29}$$

We begin by proving several convergence results related to the learning rates.

**Lemma 8** (Learning Rates). *With $\tau_n$ defined in* (29) *we have,*

$$\tau_n = \begin{cases} \mathcal{O}(n^{1-b}) & \text{if} \quad \frac{4}{5} < b < 1, \\ \mathcal{O}(\log n) & \text{if} \quad b = 1. \end{cases} \tag{30}$$

*This further implies,*

$$\sup_n \sum_{k=1}^{n} \alpha_k^2 \tau_k < \infty, \tag{31}$$

$$\sup_n \sum_{k=1}^{n} \alpha_k^2 \tau_k^2 < \infty, \tag{32}$$

$$\sup_n \sum_{k=0}^{n-1} |\alpha_k - \alpha_{k+1}| \tau_k < \infty, \tag{33}$$

$$\sup_n \sum_{k=1}^{n} \alpha_k^2 \sum_{j=1}^{i-1} \alpha_j \tau_j < \infty, \tag{34}$$

$$\sup_n \sum_{k=1}^{n} \alpha_k \sqrt{\sum_{j=1}^{k-1} \alpha_{j,k-1}^2 \tau_{j-1}^2} < \infty, \tag{35}$$

Since this Lemma is comprised of several short proofs regarding the deterministic learning rates defined in Assumption 4.4, we will decompose each result into subsections. Recall that $\alpha_n \doteq \frac{1}{(n+1)^b}$ where $\frac{4}{5} < b \leq 1$.

(30)**:**

**Proof** From the definition of $\tau_n$ in (29), we have

$$\tau_n \doteq \sum_{k=1}^{n} \alpha_k (1 - \alpha_k) \leq \sum_{k=1}^{n} \alpha_k = \sum_{k=1}^{n} \frac{1}{(k+1)^b}.$$

Case 1: $b = 1$. It is easy to see $\tau_n = \mathcal{O}(\log n)$.

Case 2: When $b < 1$, we can approximate the sum with an integral, with

$$\sum_{k=1}^{n} \frac{1}{(k+1)^b} \leq \int_1^n \frac{1}{k^b} \, dk = \frac{n^{1-b} - 1}{1 - b}$$

Therefore we have $\tau_n = \mathcal{O}(n^{1-b})$ when $b < 1$.    ∎

In analyzing the subsequent equations, we will use the fact that $\tau_n = \mathcal{O}(\log n)$ when $b = 1$ and $\tau_n = \mathcal{O}(n^{1-b})$ when $\frac{4}{5} < b < 1$. Additionally, we have $\alpha_n = \left(\frac{1}{n^b}\right)$.

(31):

**Proof** We have an order-wise approximation of the sum

$$\sum_{k=1}^{n} \alpha_k^2 \tau_k = \begin{cases} \mathcal{O}\left(\sum_{k=1}^{n} \dfrac{1}{k^{3b-1}}\right) & \text{if } \dfrac{4}{5} < b < 1, \\[3ex] \mathcal{O}\left(\sum_{k=1}^{n} \dfrac{\log(k)}{k^2}\right) & \text{if } b = 1. \end{cases}$$

In both cases of $b = 1$ and $\frac{4}{5} < b < 1$, the series clearly converge as $n \to \infty$. ∎

(32):

**Proof** This proof closely resembles that of (31). We can give an order-wise approximation of the sum

$$\sum_{k=1}^{n} \alpha_k^2 \tau_k^2 = \begin{cases} \mathcal{O}\left(\sum_{k=1}^{n} \dfrac{1}{k^{4b-2}}\right) & \text{if } \dfrac{4}{5} < b < 1, \\[3ex] \mathcal{O}\left(\sum_{k=1}^{n} \dfrac{\log^2(k)}{k^2}\right) & \text{if } b = 1. \end{cases}$$

In both cases of $b = 1$ and $\frac{4}{5} < b < 1$, the series clearly converge as $n \to \infty$. ∎

(33):

**Proof** Since $\alpha_n$ is strictly decreasing, we have $|\alpha_k - \alpha_{k+1}| = \alpha_k - \alpha_{k+1}$.

**Case 1:** For the case where $b = 1$, it is trivial to see that,

$$\sum_{k=1}^{n} |\alpha_k - \alpha_{k+1}| \tau_k = \mathcal{O}\left(\sum_{k=1}^{n} \frac{\log(k)}{k^2 + k}\right).$$

This series clearly converges.

**Case 2:** For the case where $\frac{4}{5} < b < 1$, we have

$$\alpha_n - \alpha_{n+1} = \mathcal{O}\left(\frac{1}{n^b} - \frac{1}{(n+1)^b}\right),$$
$$= \mathcal{O}\left(\frac{(n+1)^b - n^b}{n^b(n+1)^b}\right). \tag{36}$$

To analyze the behavior of this term for large $n$ we first consider the binomial expansion of $(n+1)^b$,

$$(n+1)^b = n^b\left(1 + \frac{1}{n}\right)^b = n^b(1 + b\frac{1}{n} + \frac{b(b-1)}{2}\frac{1}{n^2} + \dots)$$

Subtracting $n^b$ from $(n+1)^b$:

$$(n+1)^b - n^b = n^b(1 + b\frac{1}{n} + \frac{b(b-1)}{2}\frac{1}{n^2} + \dots) - n^b = \mathcal{O}(bn^{b-1}).$$

The leading order of the denominator of (36) is clearly $n^{2b}$, which gives

$$\alpha_n - \alpha_{n+1} = \mathcal{O}\left(\frac{bn^{b-1}}{n^{2b}}\right) = \mathcal{O}\left(\frac{b}{n^{b+1}}\right).$$

Therefore with $\tau_n = \mathcal{O}(n^{1-b})$,

$$\sum_{k=1}^{n} |\alpha_k - \alpha_{k+1}| \tau_k = \mathcal{O}\left( b \sum_{k=1}^{n} \frac{1}{k^{2b}} \right)$$

which clearly converges as $n \to \infty$ for $\frac{4}{5} < b < 1$. ∎

(34):

**Proof**

**Case 1:** In the proof for (30) we prove that $\sum_{k=1}^{n} \alpha_k = \mathcal{O}(\log n)$ when $b = 1$. Then since $\tau_k$ is increasing, we have

$$\sum_{k=1}^{n} \alpha_k^2 \sum_{j=1}^{k-1} \alpha_j \tau_j \leq \sum_{k=1}^{n} \alpha_k^2 \tau_k \sum_{j=1}^{k-1} \alpha_j = \mathcal{O}\left( \sum_{k=1}^{n} \frac{\log^2 k}{k^2} \right),$$

∎

which clearly converges as $n \to \infty$.

**Case 2:** For the case when $b \in (\frac{4}{5}, 1]$, we first consider the inner sum of (34),

$$\sum_{j=1}^{k-1} \alpha_j \tau_j = \mathcal{O}\left( \sum_{j=1}^{k-1} \frac{1}{j^{2b-1}} \right),$$

which we can approximate by an integral,

$$\int_{1}^{k} \frac{1}{x^{2b-1}}\, dx = \mathcal{O}(k^{2-2b}).$$

Therefore,

$$\sum_{k=1}^{n} \alpha_k^2 \sum_{j=1}^{k-1} \alpha_j \tau_j = \mathcal{O}\left( \sum_{k=1}^{n} \frac{k^{2-2b}}{k^{2b}} \right) = \mathcal{O}\left( \sum_{k=1}^{n} \frac{1}{k^{4b-2}} \right),$$

which converges for $\frac{4}{5} < b \leq 1$ as $n \to \infty$.

(35): **Proof**

**Case 1:** For $b = 1$, because we have $\alpha_{j,i} < \alpha_{j+1,i}$ and $\alpha_{i,i} = \alpha_i$ from Lemma 5, we have the order-wise approximation,

$$\sum_{i=1}^{n} \alpha_i \sqrt{\sum_{j=1}^{i-1} \alpha_{j,i-1}^2 \tau_{j-1}^2} \leq \sum_{i=1}^{n} \alpha_i \sqrt{\alpha_{i-1}^2 \tau_{i-1}^2 \sum_{j=1}^{i-1} 1}, \qquad (\tau_i \text{ is increasing})$$

$$= \sum_{i=1}^{n} \alpha_i \alpha_{i-1} \tau_{i-1} \sqrt{i-1}.$$

$$= \mathcal{O}\left( \sum_{i=1}^{n} \frac{\log(i-1)}{i\sqrt{(i-1)}} \right)$$

$$= \mathcal{O}\left( \sum_{i=1}^{n} \frac{\log(i-1)}{i^{3/2}} \right),$$

which clearly converges.

**Case 2:**   We have,

$$\sum_{i=1}^{n} \alpha_i \sqrt{\sum_{j=1}^{i-1} \alpha_{j,i-1}^2 \tau_{j-1}^2} \le \sum_{i=1}^{n} \alpha_i \tau_{i-1} \sqrt{\sum_{j=1}^{i-1} \alpha_{j,i-1}^2}, \qquad (\tau_i \text{ is increasing})$$

$$= \sum_{i=1}^{n} \alpha_i \tau_{i-1} \sqrt{\alpha_i}. \qquad (\text{Lemma } 6)$$

$$= \mathcal{O}\left(\sum_{i=1}^{n} \frac{i^{1-b}}{i^b \sqrt{i^b}}\right)$$

$$= \mathcal{O}\left(\sum_{i=1}^{n} \frac{1}{i^{5b/2-1}}\right),$$

which converges for $\frac{4}{5} < b < 1$. ■

Then, under Assumption 4.5, we prove additional results about the convergence of the first and second moments of the additive noise $\left\{\epsilon_n^{(1)}\right\}$.

**Lemma 9.** *Let Assumptions 4.4 and 4.5 hold. Then, we have*

$$\mathbb{E}\left[\left\|\epsilon_n^{(1)}\right\|_2\right] = \mathcal{O}\left(\frac{1}{\sqrt{n}}\right), \tag{37}$$

$$\sup_n \sum_{k=1}^{n} \alpha_k \mathbb{E}\left[\left\|\epsilon_k^{(1)}\right\|\right] < \infty, \tag{38}$$

$$\sup_n \sum_{k=1}^{n} \alpha_k \mathbb{E}\left[\left\|\epsilon_k^{(1)}\right\|^2\right] < \infty, \tag{39}$$

$$\sup_n \sum_{k=1}^{n} \alpha_k^2 \mathbb{E}\left[\left\|\epsilon_k^{(1)}\right\|^2\right] < \infty, \tag{40}$$

$$\sup_n \sum_{k=1}^{n} \alpha_k \sum_{j=1}^{k-1} \alpha_{j,k-1} \mathbb{E}\left[\left\|\epsilon_j^{(1)}\right\|\right] < \infty. \tag{41}$$

**Proof**   Recall that by Assumption 4.5 we have $\mathbb{E}\left[\left\|\epsilon_n^{(1)}\right\|^2\right] = \mathcal{O}\left(\frac{1}{n}\right)$. Also recall that $\alpha_k = \mathcal{O}\left(\frac{1}{n^b}\right)$ with $\frac{4}{5} < b \le 1$. Then, we can prove the following equations:

(37):   By Jensen's inequality, we have

$$\mathbb{E}\left[\left\|\epsilon_n^{(1)}\right\|\right] \le \sqrt{\mathbb{E}\left[\left\|\epsilon_n^{(1)}\right\|^2\right]} = \mathcal{O}\left(\frac{1}{\sqrt{n}}\right).$$

(38):

$$\sum_{k=1}^{n} \alpha_k \mathbb{E}\left[\left\|\epsilon_k^{(1)}\right\|\right] = \mathcal{O}\left(\sum_{k=1}^{n} \frac{1}{k^{b+\frac{1}{2}}}\right)$$

which clearly converges as $n \to \infty$ for $\frac{4}{5} < b \le 1$.

(39):

$$\sum_{k=1}^{n} \alpha_k \mathbb{E}\left[\left\|\epsilon_k^{(1)}\right\|^2\right] = \mathcal{O}\left(\sum_{k=1}^{n} \frac{1}{k^{b+1}}\right)$$

which clearly converges as $n \to \infty$ for $\frac{4}{5} < b \le 1$.

(40):

$$\sum_{k=1}^{n} \alpha_k^2 \mathbb{E}\left[\left\|\epsilon_k^{(1)}\right\|^2\right] = \mathcal{O}\left(\sum_{k=1}^{n} \frac{1}{k^{2b+1}}\right)$$

which clearly converges as $n \to \infty$ for $\frac{4}{5} < b \leq 1$.

(41):

$$\sum_{k=1}^{n} \alpha_k \sum_{j=1}^{k-1} \alpha_{j,k-1} \mathbb{E}\left[\left\|\epsilon_j^{(1)}\right\|\right] \leq \sum_{k=1}^{n} \alpha_k^2 \sum_{j=1}^{k-1} \mathbb{E}\left[\left\|\epsilon_j^{(1)}\right\|\right], \qquad \text{(Lemma 5)}$$

$$= \mathcal{O}\left(\sum_{k=1}^{n} \frac{1}{k^{2b}} \sum_{j=1}^{k-1} \frac{1}{\sqrt{j}}\right). \qquad \text{(Lemma 9)}$$

It can be easily verified with an integral approximation that $\sum_{j=1}^{k-1} \frac{1}{\sqrt{j}} = \mathcal{O}(\sqrt{k})$. This further implies

$$\sum_{k=1}^{n} \alpha_k \sum_{j=1}^{k-1} \alpha_{j,k-1} \mathbb{E}\left[\left\|\epsilon_j^{(1)}\right\|\right] = \mathcal{O}\left(\sum_{k=1}^{n} \frac{1}{k^{2b-\frac{1}{2}}}\right),$$

which converges as $n \to \infty$ for $\frac{4}{5} < b \leq 1$. ∎

Next, in Lemma 10, we upper-bound the iterates $\{x_n\}$.

**Lemma 10.** *For each $\{x_n\}$, we have*

$$\|x_n\| \leq \|x_0\| + C_H \sum_{k=1}^{n} \alpha_k + \sum_{k=1}^{n} \alpha_k \left\|\epsilon_k^{(1)}\right\| \leq C_{10}\tau_n + \sum_{k=1}^{n} \alpha_k \left\|\epsilon_k^{(1)}\right\|,$$

*where $C_{10}$ is a deterministic constant.*

**Proof** Applying $\|\cdot\|$ to both sides of (SKM with Markovian and Additive Noise) gives,

$$\|x_{n+1}\| = \left\|(1 - \alpha_{n+1})x_n + \alpha_{n+1}\left(H(x_n, Y_{n+1}) + \epsilon_{n+1}^{(1)}\right)\right\|,$$

$$\leq (1 - \alpha_{n+1})\|x_n\| + \alpha_{n+1}\|H(x_n, Y_{n+1})\| + \alpha_{n+1}\left\|\epsilon_{n+1}^{(1)}\right\|,$$

$$\leq (1 - \alpha_{n+1})\|x_n\| + \alpha_{n+1}(C_H + \|x_n\|) + \alpha_{n+1}\left\|\epsilon_{n+1}^{(1)}\right\|, \qquad \text{(By (12))}$$

$$= \|x_n\| + \alpha_{n+1}C_H + \alpha_{n+1}\left\|\epsilon_{n+1}^{(1)}\right\|.$$

A simple induction shows that almost surely,

$$\|x_n\| \leq \|x_0\| + C_H \sum_{k=1}^{n} \alpha_k + \sum_{k=1}^{n} \alpha_k \left\|\epsilon_k^{(1)}\right\|.$$

Since $\{\alpha_n\}$ is monotonically decreasing, we have

$$\|x_n\| \leq \|x_0\| + \frac{C_H}{(1 - \alpha_1)} \sum_{k=1}^{n} \alpha_k(1 - \alpha_k) + \sum_{k=1}^{n} \alpha_k \left\|\epsilon_k^{(1)}\right\|,$$

$$= \|x_0\| + \frac{C_H}{(1 - \alpha_1)}\tau_n + \sum_{k=1}^{n} \alpha_k \left\|\epsilon_k^{(1)}\right\|,$$

$$\leq \max\left\{\|x_0\|, \frac{C_H}{(1 - \alpha_1)}\right\}(1 + \tau_n) + \sum_{k=1}^{n} \alpha_k \left\|\epsilon_k^{(1)}\right\|.$$

Therefore, since $\tau_n$ is monotonically increasing, there exists some constant we denote as $C_{10}$ such that

$$\|x_n\| \leq C_{10}\tau_n + \sum_{k=1}^{n} \alpha_k \left\|\epsilon_k^{(1)}\right\|.$$

∎

**Lemma 11.** *With $\nu(x, y)$ as defined in (16), we have*

$$\|\nu(x, y) - \nu(x', y)\| \leq C_{11}\|x - x'\|, \tag{42}$$

*which further implies*

$$\|\nu(x, y)\| \leq C_{11}(C'_{11} + \|x\|),$$

*where $C_{11}, C'_{11}$ are deterministic constants.*

**Proof** Since we work with a finite $\mathcal{Y}$, we will use functions and matrices interchangeably. For example, given a function $f : \mathcal{Y} \to \mathbb{R}^d$, we also use $f$ to denote a matrix in $\mathbb{R}^{(|\mathcal{Y}| \times d)}$ whose $y$-th row is $f(y)^\top$. Similarly, a matrix in $\mathbb{R}^{(|\mathcal{Y}| \times d)}$ also corresponds to a function $\mathcal{Y} \to \mathbb{R}^d$.

Let $\nu_x \in \mathbb{R}^{|\mathcal{Y}| \times d}$ denote the function $y \mapsto \nu(x, y)$ and let $H_x \in \mathbb{R}^{|\mathcal{Y}| \times d}$ denote the function $y \mapsto H(x, y)$. Theorem 8.2.6 of Puterman (2014) then ensures that

$$\nu_x = H_{\mathcal{Y}} H_x,$$

where $H_{\mathcal{Y}} \in \mathbb{R}^{|\mathcal{Y}| \times |\mathcal{Y}|}$ is the fundamental matrix of the Markov chain depending only on the chain's transition matrix $P$. The exact expression of $H_{\mathcal{Y}}$ is inconsequential and we refer the reader to Puterman (2014) for details. Then we have for any $i = 1, \ldots, d$,

$$\nu_x[y, i] = \sum_{y'} H_{\mathcal{Y}}[y, y'] H_x[y', i].$$

This implies that

$$|\nu_x[y, i] - \nu_{x'}[y, i]| \leq \sum_{y'} H_{\mathcal{Y}}[y, y'] |H_x[y', i] - H_{x'}[y', i]|$$

$$\leq \sum_{y'} H_{\mathcal{Y}}[y, y'] \|H(x, y) - H(x', y')\|_\infty$$

$$\leq \sum_{y'} H_{\mathcal{Y}}[y, y'] \|x - x'\|_\infty \qquad \text{(Assumption 4.2)}$$

$$\leq \|H_{\mathcal{Y}}\|_\infty \|x - x'\|_\infty,$$

yielding

$$\|\nu(x, y) - \nu(x', y)\|_\infty \leq \|H_{\mathcal{Y}}\|_\infty \|x - x'\|_\infty.$$

The equivalence between norms in finite dimensional space ensures that there exists some $C_{11}$ such that (42) holds. Letting $x' = 0$ then yields

$$\|\nu(x, y)\| \leq C_{11}(\|\nu(0, y)\| + \|x\|).$$

Define $C'_{11} \doteq \max_y \|\nu(0, y)\|$, we get

$$\|\nu(x, y)\| \leq C_{11}(C'_{11} + \|x\|).$$

∎

**Lemma 12.** *We have for any $y \in \mathcal{Y}$,*

$$\|\nu(x_n, y)\| \leq \zeta_{12}\tau_n,$$

*where $\zeta$ is a possibly sample-path dependent constant. Additionally, we have*

$$\mathbb{E}[\|\nu(x_n, y)\|] \leq C_{12}\tau_n,$$

*where $C_{12}$ is a deterministic constant.*

**Proof** Having proven that $\nu(x, y)$ is Lipschitz continuous in $x$ in Lemma 11, we have

$$\|\nu(x_n, y)\| \leq C_{11}(C'_{11} + \|x_n\|), \qquad \text{(Lemma 11)}$$

$$\leq C_{11}\left(C'_{11} + C_{10}\tau_n + \sum_{k=1}^{n}\alpha_k\left\|\epsilon_k^{(1)}\right\|\right). \qquad \text{(Lemma 10)}$$

$$= \mathcal{O}\left(\tau_n + \sum_{k=1}^{n}\alpha_k\left\|\epsilon_k^{(1)}\right\|\right).$$

Since (14) in Assumption 4.5 assures us that $\sum_{k=1}^{\infty}\alpha_k\left\|\epsilon_k^{(1)}\right\|$ is finite almost surely while $\tau_n$ is monotonically increasing, then there exists some possibly sample-path dependent constant $\zeta_{12}$ such that

$$\|\nu(x_n, y)\| \leq \zeta_{12}\tau_n.$$

We can also prove a deterministic bound on the expectation of $\|\nu(x_n, Y_{n+1})\|$,

$$\mathbb{E}[\|\nu(x_n, y)\|] = \mathcal{O}\left(\mathbb{E}\left[\tau_n + \sum_{k=1}^{n}\alpha_k\left\|\epsilon_k^{(1)}\right\|\right]\right),$$

$$= \mathcal{O}\left(\tau_n + \sum_{k=1}^{n}\alpha_k\mathbb{E}\left[\left\|\epsilon_k^{(1)}\right\|\right]\right).$$

By Lemma 9, its easy to see that $\sum_{k=1}^{n}\alpha_k\mathbb{E}\left[\left\|\epsilon_k^{(1)}\right\|\right] < \infty$. Therefore, there exists some deterministic constant $C_{12}$ such that

$$\mathbb{E}[\|\nu(x_n, y)\|] \leq C_{12}\tau_n.$$

∎

Although the two statements in Lemma 12 appear similar, their difference is crucial. Assumption 4.5 and (14) only ensure the existence of a sample-path dependent constant $\zeta_{12}$ but its form is unknown, preventing its use for expectations or explicit bounds. In contrast, using (15) from Assumption 4.5, we derive a universal constant $C_{12}$.

**Lemma 13.** *For each $\{M_n\}$, defined in (18), we have*

$$\|M_{n+1}\| \leq \zeta_{13}\tau_n,$$

*where $\zeta_{13}$ is a the sample-path dependent constant.*

**Proof** Applying $\|\cdot\|$ to (18) gives

$$\|M_{n+1}\| = \|\nu(x_n, Y_{n+2}) - P\nu(x_n, Y_{n+1})\|,$$

$$\leq \|P\nu(x_n, Y_{n+1})\| + \|\nu(x_n, Y_{n+2})\|,$$

$$= \left\|\sum_{y'\in\mathcal{Y}}P(Y_{n+1}, y')\nu(x_n, y')\right\| + \|\nu(x_n, Y_{n+2})\|,$$

$$\leq \sum_{y'\in\mathcal{Y}}\|P(Y_{n+1}, y')\nu(x_n, y')\| + \|\nu(x_n, Y_{n+2})\|,$$

$$= \left(\max_{y\in\mathcal{Y}}\|\nu(x_n, y)\|\right)\sum_{y'\in\mathcal{Y}}|P(Y_{n+1}, y')| + \|\nu(x_n, Y_{n+2})\|,$$

$$\leq 2\max_{y\in\mathcal{Y}}\|\nu(x_n, y)\| \qquad (43)$$

Under Assumption 4.5, we can apply the sample-path dependent bound from Lemma 12,

$$\|M_{n+1}\| \leq 2\zeta_{12}\tau_n, \qquad \text{(Lemma 12)}$$

$$= \zeta_{13}\tau_n,$$

with $\zeta_{13} \doteq 2\zeta_{12}$. ∎

**Lemma 14.** *For each $\{M_n\}$, defined in* (18)*, we have*

$$\mathbb{E}\left[\|M_{n+1}\|^2 \mid \mathcal{F}_{n+1}\right] \leq C_{14}'(1 + \|x_n\|^2), \tag{44}$$

*and*

$$\mathbb{E}\left[\|M_{n+1}\|_2^2\right] \leq C_{14}^2 \tau_n^2, \tag{45}$$

*where $C_{14}'$ and $C_{14}$ are deterministic constants and*

$$\mathcal{F}_{n+1} \doteq \sigma(x_0, Y_1, \ldots, Y_{n+1})$$

*is the $\sigma$-algebra until time $n + 1$.*

**Proof** First, to prove (44), we have

$$\mathbb{E}\left[\|M_{n+1}\|^2 \mid \mathcal{F}_{n+1}\right] \leq 4 \max_{y \in \mathcal{Y}} \|\nu(x_n, y)\|^2 = \mathcal{O}\left(1 + \|x_n\|^2\right),$$

where the first inequality results form (43) in Lemma 13 and the second inequality results from Lemma 11.

Then, to prove (45), from Lemma 10 we then have,

$$\mathbb{E}\left[\|\nu(x_n, y)\|^2\right] \leq \mathbb{E}\left[1 + \left(C_{10}\tau_n + \sum_{k=1}^{n} \alpha_k \left\|\epsilon_k^{(1)}\right\|\right)^2\right] = \mathcal{O}\left(\tau_n^2 + \mathbb{E}\left[\left(\sum_{k=1}^{n} \alpha_k \left\|\epsilon_k^{(1)}\right\|\right)^2\right]\right).$$

Recall that by Assumption 4.5, $\mathbb{E}\left[\left\|\epsilon_k^{(1)}\right\|^2\right] = \mathcal{O}\left(\frac{1}{k}\right)$. Examining the right-most term we then have,

$$\mathbb{E}\left[\left(\sum_{k=1}^{n} \alpha_k \left\|\epsilon_k^{(1)}\right\|\right)^2\right] \leq \mathbb{E}\left[\left(\sum_{k=1}^{n} \alpha_k\right)\left(\sum_{k=1}^{n} \alpha_k \left\|\epsilon_k^{(1)}\right\|^2\right)\right] \qquad \text{(Cauchy-Schwarz)}$$

$$= \mathcal{O}\left(\sum_{k=1}^{n} \alpha_k\right) \qquad \text{(By (39) in Lemma 9)}$$

$$= \mathcal{O}\left(\frac{1}{1 - \alpha_1} \sum_{k=1}^{n} \alpha_k(1 - \alpha_1)\right) = \mathcal{O}\left(\sum_{k=1}^{n} \alpha_k(1 - \alpha_k)\right)$$

$$= \mathcal{O}(\tau_n).$$

We then have

$$\mathbb{E}\left[\|\nu(x_n, y)\|^2\right] = \mathcal{O}(\tau_n^2). \tag{46}$$

Because our bound on $\mathbb{E}\left[\|\nu(x_n, y)\|^2\right]$ is independent of $y$, we have

$$\mathbb{E}\left[\|M_{n+1}\|^2\right] = \mathcal{O}\left(\mathbb{E}\left[\|\nu(x_n, y)\|^2\right]\right) = \mathcal{O}(\tau_n^2). \qquad \text{(By (46))}$$

Due to the equivalence of norms in finite-dimensional spaces, there exists a deterministic constant $C_{14}$ such that (45) holds. ■

Now, we are ready to present four additional lemmas which we will use to bound the four noise terms in (27).

**Lemma 15.** *With $\left\{\overline{\overline{M}}_n\right\}$ defined in* (27)*,*

$$\lim_{n \to \infty} \overline{\overline{M}}_n < \infty, \quad a.s.$$

**Proof** We first observe that the sequence $\left\{\overline{\overline{M}}_n\right\}$ defined in (27) is positive and monotonically increasing. Therefore by the monotone convergence theorem, it converges almost surely to a (possibly infinite) limit which we denote as,

$$\overline{\overline{M}}_\infty \doteq \lim_{n\to\infty} \overline{\overline{M}}_n \quad \text{a.s.}$$

Then, we will utilize a generalization of Lebesgue's monotone convergence theorem (Lemma 7) to prove that the limit $\overline{\overline{M}}_\infty$ is finite almost surely. From Lemma 7, we see that

$$\mathbb{E}\left[\overline{\overline{M}}_\infty\right] = \lim_{n\to\infty} \mathbb{E}\left[\overline{\overline{M}}_n\right].$$

Therefore, to prove that $\overline{\overline{M}}_\infty$ is almost surely finite, it is sufficient to prove that $\lim_{n\to\infty} \mathbb{E}\left[\overline{\overline{M}}_n\right] < \infty$. To this end, we proceed by bounding the expectation of $\left\{\overline{\overline{M}}_n\right\}$, by first starting with $\left\{\overline{M}_n\right\}$ from (26). We have,

$$\mathbb{E}\left[\left\|\overline{M}_n\right\|\right] = \mathbb{E}\left[\left\|\sum_{i=1}^n \alpha_{i,n} M_i\right\|\right],$$

$$= \mathcal{O}\left(\sqrt{\mathbb{E}\left[\left\|\sum_{i=1}^n \alpha_{i,n} M_i\right\|_2^2\right]}\right), \qquad \text{(Jensen's Ineq.)}$$

$$= \mathcal{O}\left(\sqrt{\sum_{i=1}^n \alpha_{i,n}^2 \mathbb{E}\left[\|M_i\|_2^2\right]}\right), \qquad (M_i \text{ is a Martingale Difference Series})$$

$$= \mathcal{O}\left(\sqrt{\sum_{i=1}^n \alpha_{i,n}^2 \tau_i^2}\right), \qquad \text{(Lemma 14)}$$

Then using the definition of $\left\{\overline{\overline{M}}_n\right\}$ from (27), we have

$$\mathbb{E}\left[\overline{\overline{M}}_n\right] = \sum_{i=1}^n \alpha_i \mathbb{E}\left[\left\|\overline{M}_{i-1}\right\|\right] = \mathcal{O}\left(\sum_{i=1}^n \alpha_i \sqrt{\sum_{j=1}^{i-1} \alpha_{j,i-1}^2 \tau_{j-1}^2}\right).$$

Then, by (35) in Lemma 8, we have

$$\sup_n \mathbb{E}\left[\overline{\overline{M}}_n\right] < \infty,$$

and since $\left\{\mathbb{E}\left[\overline{\overline{M}}_n\right]\right\}$ is also monotonically increasing, we have

$$\lim_{n\to\infty} \mathbb{E}\left[\overline{\overline{M}}_n\right] < \infty,$$

which implies that $\overline{\overline{M}}_\infty < \infty$ almost surely. ∎

**Lemma 16.** *With $\left\{\overline{\overline{\epsilon}}_n^{(1)}\right\}$ defined in (27),*

$$\lim_{n\to\infty} \overline{\overline{\epsilon}}_n^{(1)} < \infty, \quad \text{a.s.}$$

**Proof** We first observe that the sequence $\left\{\overline{\overline{\epsilon}}_n^{(1)}\right\}$ defined in (27) is positive and monotonically increasing. Therefore by the monotone convergence theorem, it converges almost surely to a (possibly infinite) limit which we denote as,

$$\overline{\overline{\epsilon}}_\infty^{(1)} \doteq \lim_{n\to\infty} \overline{\overline{\epsilon}}_n^{(1)} \quad \text{a.s.}$$

Then, we utilize a generalization of Lebesgue's monotone convergence theorem (Lemma 7) to prove that the limit $\bar{\bar{\epsilon}}_\infty^{(1)}$ is finite almost surely. By Lemma 7, we have

$$\mathbb{E}\left[\bar{\bar{\epsilon}}_\infty^{(1)}\right] = \lim_{n\to\infty} \mathbb{E}\left[\bar{\bar{\epsilon}}_n^{(1)}\right].$$

Therefore, to prove that $\bar{\bar{\epsilon}}_\infty^{(1)}$ is almost surely finite, it is sufficient to prove that $\lim_{n\to\infty} \mathbb{E}\left[\bar{\bar{\epsilon}}_n^{(1)}\right] < \infty$. To this end, we proceed by bounding the expectation of $\left\{\bar{\bar{\epsilon}}_n^{(1)}\right\}$,

$$\mathbb{E}\left[\bar{\bar{\epsilon}}_n^{(1)}\right] = \sum_{i=1}^n \alpha_i \mathbb{E}\left[\left\|\bar{\epsilon}_{i-1}^{(1)}\right\|\right] \le \sum_{i=1}^n \alpha_i \sum_{j=1}^{i-1} \alpha_{j,i-1} \mathbb{E}\left[\left\|\epsilon_j^{(1)}\right\|\right].$$

Then, by (41) in Lemma 9, we have,

$$\sup_n \mathbb{E}\left[\bar{\bar{\epsilon}}_n^{(1)}\right] < \infty,$$

and since $\left\{\mathbb{E}\left[\bar{\bar{\epsilon}}_n^{(1)}\right]\right\}$ is also monotonically increasing, we have

$$\lim_{n\to\infty} \mathbb{E}\left[\bar{\bar{\epsilon}}_n^{(1)}\right] < \infty.$$

which implies that $\bar{\bar{\epsilon}}_\infty^{(1)} < \infty$ almost surely.

■

**Lemma 17.** *With $\left\{\bar{\bar{\epsilon}}_n^{(2)}\right\}$ defined in (27), we have*

$$\lim_{n\to\infty} \bar{\bar{\epsilon}}_n^{(2)} < \infty \quad \text{a.s.}$$

**Proof** Starting with the definition of $\bar{\epsilon}_n^{(2)}$ from (26), we have,

$$\bar{\epsilon}_n^{(2)} = \sum_{i=1}^n \alpha_{i,n} \epsilon_i^{(2)}$$

$$= -\sum_{i=1}^n \alpha_{i,n}(\nu(x_i, Y_{i+1}) - \nu(x_{i-1}, Y_i)),$$

$$= -\sum_{i=1}^n \alpha_{i,n}\nu(x_i, Y_{i+1}) - \alpha_{i-1,n}\nu(x_{i-1}, Y_i) + \alpha_{i-1,n}\nu(x_{i-1}, Y_i) - \alpha_{i,n}\nu(x_{i-1}, Y_i),$$

$$= -\alpha_{n,n}\nu(x_n, Y_{n+1}) - \sum_{i=1}^n (\alpha_{i-1,n} - \alpha_{i,n})\nu(x_{i-1}, Y_i). \qquad (\alpha_0 \doteq 0)$$

Since we have $\alpha_{n,n} = \alpha_n$ by definition, the triangle inequality gives

$$\left\|\bar{\epsilon}_n^{(2)}\right\| \le \alpha_n \|\nu(x_n, Y_{n+1})\| + \sum_{i=1}^n |\alpha_{i-1,n} - \alpha_{i,n}| \,\|\nu(x_{i-1}, Y_i)\|,$$

$$\le \zeta_{12}\left(\alpha_n \tau_n + \sum_{i=1}^n |\alpha_{i-1,n} - \alpha_{i,n}| \,\tau_{i-1}\right), \qquad \text{(Lemma 12)}$$

$$\le \zeta_{12}\left(\alpha_n \tau_n + \tau_n \sum_{i=1}^n (\alpha_{i,n} - \alpha_{i-1,n})\right), \qquad \text{(Lemma 5)}$$

$$\le 2\zeta_{12}\alpha_n \tau_n. \qquad (\alpha_0 \doteq 0)$$

Therefore, there exists a sample-path dependent constant we denote as $\zeta_{17}$ such that

$$\left\| \bar{\epsilon}_n^{(2)} \right\| \leq \zeta_{17} \alpha_n \tau_n.$$

Therefore, from the definition of $\bar{\bar{\epsilon}}_n^{(2)}$ in (23), we have

$$\bar{\bar{\epsilon}}_n^{(2)} = \sum_{i=1}^{n} \alpha_i \left\| \bar{\epsilon}_{i-1}^{(2)} \right\|,$$

$$\leq \zeta_{17} \sum_{i=1}^{n} \alpha_i \alpha_{i-1} \tau_{i-1},$$

$$= \zeta_{17} \sum_{k=1}^{n-1} \alpha_{k+1} \alpha_k \tau_k, \qquad\qquad (\alpha_0 \doteq 0)$$

$$\leq \zeta_{17} \sum_{k=1}^{n} \alpha_k^2 \tau_k, \qquad\qquad (\alpha_k \text{ is decreasing for } k \geq 1)$$

which is almost surely finite by Lemma 8. ∎

**Lemma 18.** *With $\left\{ \bar{\bar{\epsilon}}_n^{(3)} \right\}$ defined in (27), we have*

$$\lim_{n \to \infty} \bar{\bar{\epsilon}}_n^{(3)} < \infty, \quad a.s.$$

**Proof** Beginning with the definition of $\bar{\epsilon}_n^{(3)}$ in (26), we have

$$\left\| \bar{\epsilon}_n^{(3)} \right\| = \left\| \sum_{i=1}^{n} \alpha_{i,n} (\nu(x_i, Y_{i+1}) - \nu(x_{i-1}, Y_{i+1})) \right\|,$$

$$\leq \sum_{i=1}^{n} \alpha_{i,n} \| \nu(x_i, Y_{i+1}) - \nu(x_{i-1}, Y_{i+1}) \|,$$

$$\leq C_{11} \sum_{i=1}^{n} \alpha_{i,n} \| x_i - x_{i-1} \|, \qquad\qquad (\text{Lemma } 11)$$

$$\leq C_{11} \sum_{i=1}^{n} \alpha_{i,n} \alpha_i \left( \| H(x_{i-1}, Y_i) \| + \| x_{i-1} \| + \left\| \epsilon_i^{(1)} \right\| \right), \qquad (\text{By (SKM with Markovian and Additive Noise)})$$

$$\leq C_{11} \sum_{i=1}^{n} \alpha_{i,n} \alpha_i \left( 2 \| x_{i-1} \| + C_H + \left\| \epsilon_i^{(1)} \right\| \right), \qquad\qquad (\text{By (12)})$$

$$\leq C_{11} \sum_{i=1}^{n} \alpha_{i,n} \alpha_i \left( 2 C_{10} \tau_{i-1} + 2 \sum_{k=1}^{i-1} \alpha_k \left\| \epsilon_k^{(1)} \right\| + C_H + \left\| \epsilon_i^{(1)} \right\| \right), \quad (\text{Lemma } 10)$$

Because Assumption 4.5 assures us that $\sum_{k=1}^{\infty} \alpha_k \left\| \epsilon_k^{(1)} \right\|$ is almost surely finite, then there exists some sample-path dependent constant we denote as $\zeta_{18}$ where,

$$\left\| \bar{\epsilon}_n^{(3)} \right\| \leq \zeta_{18} \sum_{i=1}^{n} \alpha_{i,n} \alpha_i \left( \tau_{i-1} + \left\| \epsilon_i^{(1)} \right\| \right), \qquad\qquad (\text{Assumption } 4.5)$$

$$\leq \zeta_{18} \left( \sum_{i=1}^{n} \alpha_{i,n} \alpha_i \tau_i + \sum_{i=1}^{n} \alpha_{i,n} \alpha_i \left\| \epsilon_i^{(1)} \right\| \right), \qquad\qquad (\tau_i \text{ is increasing})$$

$$\leq \zeta_{18} \alpha_n \left( \sum_{i=1}^{n} \alpha_i \tau_i + \sum_{i=1}^{n} \alpha_i \left\| \epsilon_i^{(1)} \right\| \right). \qquad\qquad (\text{Lemma } 5).$$

Again, from Assumption 4.5 we can conclude that there exists some other sample-path dependent constant we denote as $\zeta_{18}'$ where

$$\left\| \bar{\epsilon}_n^{(3)} \right\| \leq \zeta_{18}' \alpha_n \sum_{i=1}^{n} \alpha_i \tau_i.$$

Therefore, from the definition of $\bar{\bar{\epsilon}}_n^{(3)}$ in (23)

$$\bar{\bar{\epsilon}}_n^{(3)} \leq \zeta_{18}' \sum_{i=1}^{n} \alpha_i^2 \sum_{j=1}^{i-1} \alpha_j \tau_j.$$

So, by (34) in Lemma 8

$$\sup_n \bar{\bar{\epsilon}}_n^{(3)} \leq \sup_n \zeta_{18}' \sum_{i=1}^{n} \alpha_i^2 \sum_{j=1}^{i-1} \alpha_j \tau_j < \infty \quad \text{a.s.}$$

Then, the monotone convergence theorem proves the lemma. ∎

To prove (24) holds almost surely, we introduce four lemmas which we will subsequently use to prove an extension of Theorem 2 from (Borkar, 2009) in Section C.

**Lemma 19.** *We have*

$$\sup_n \left\| \sum_{k=1}^{n} \alpha_k M_k \right\| < \infty \quad \text{a.s.}$$

**Proof** Recall that $M_k$ is a Martingale difference series. Then, the Martingale sequence

$$\left\{ \sum_{k=1}^{n} \alpha_k M_k \right\}$$

is bounded in $L^2$ with,

$$\mathbb{E}\left[ \left\| \sum_{k=1}^{n} \alpha_k M_k \right\|_2 \right] \leq \sqrt{\mathbb{E}\left[ \left\| \sum_{k=1}^{n} \alpha_k M_k \right\|_2^2 \right]}, \qquad \text{(Jensen's Ineq.)}$$

$$= \sqrt{\sum_{k=1}^{n} \alpha_k^2 \mathbb{E}\left[ \|M_k\|_2^2 \right]}, \qquad (M_i \text{ is a Martingale Difference Series})$$

$$\leq C_{14} \sqrt{\sum_{k=1}^{n} \alpha_k^2 \tau_k^2}. \qquad \text{(Lemma 14)}$$

Lemma 8 then gives

$$\sup_n C_{14} \sqrt{\sum_{k=1}^{n} \alpha_k^2 \tau_k^2} < \infty$$

Doob's martingale convergence theorem implies that $\{\sum_{k=1}^{n} \alpha_k M_k\}$ converges to an almost surely finite random variable, which proves the lemma. ∎

**Lemma 20.** *We have,*

$$\sup_n \left\| \sum_{k=1}^{n} \alpha_k \epsilon_k^{(2)} \right\| < \infty \quad \text{a.s.}$$

**Proof** Utilizing the definition of $\epsilon_k^{(2)}$ in (19), we have

$$\sum_{k=1}^{n} \alpha_k \epsilon_k^{(2)} = -\sum_{k=1}^{n} \alpha_k(\nu(x_k, Y_{k+1}) - \nu(x_{k-1}, Y_k)),$$

$$= -\sum_{k=1}^{n} \alpha_k \nu(x_k, Y_{k+1}) - \alpha_{k-1}\nu(x_{k-1}, Y_k) + \alpha_{k-1}\nu(x_{k-1}, Y_k) - \alpha_k\nu(x_{k-1}, Y_k),$$

$$= -\alpha_n \nu(x_n, Y_{n+1}) - \sum_{k=1}^{n} (\alpha_{k-1} - \alpha_k)\nu(x_{k-1}, Y_k). \qquad (\alpha_0 = 0)$$

$$(47)$$

The triangle inequality gives

$$\left\| \sum_{k=1}^{n} \alpha_k \epsilon_k^{(2)} \right\| \le \alpha_n \|\nu(x_n, Y_{n+1})\| + \sum_{k=1}^{n} |\alpha_{k-1} - \alpha_k| \, \|\nu(x_{k-1}, Y_k)\|,$$

$$\le \zeta_{12}\left( \alpha_n \tau_n + \sum_{k=1}^{n} |\alpha_{k-1} - \alpha_k| \, \tau_{k-1} \right), \qquad \text{(Lemma 12)}$$

$$= \zeta_{12}\left( \alpha_n \tau_n + \alpha_1 \tau_1 + \sum_{k=1}^{n-1} |\alpha_k - \alpha_{k+1}| \tau_k \right) \qquad (\alpha_0 \doteq 0).$$

Its easy to see that $\lim_{n\to\infty} \alpha_n \tau_n = 0$, and $\alpha_1 \tau_1$ is simply a deterministic and finite constant. Therefore, by Lemma 8 we have

$$\sup_n \sum_{k=1}^{n} |\alpha_k - \alpha_{k+1}| \tau_k < \infty \quad \text{a.s.}$$

which proves the lemma.

∎

**Lemma 21.** *We have,*

$$\sup_n \left\| \sum_{k=1}^{n} \alpha_k \epsilon_k^{(3)} \right\| < \infty \quad \text{a.s.}$$

**Proof** Utilizing the definition of $\epsilon_k^{(3)}$ in (20), we have

$$\left\| \sum_{k=1}^{n} \alpha_k \epsilon_k^{(3)} \right\| = \left\| \sum_{k=1}^{n} \alpha_k(\nu(x_k, Y_{i+1}) - \nu(x_{k-1}, Y_{i+1})) \right\|,$$

$$\le \sum_{k=1}^{n} \alpha_k \|\nu(x_k, Y_{i+1}) - \nu(x_{k-1}, Y_{i+1})\|,$$

$$\le C_{11} \sum_{k=1}^{n} \alpha_k \|x_k - x_{k-1}\|, \qquad \text{(Lemma 11)}$$

$$\le C_{11} \sum_{k=1}^{n} \alpha_k^2 \left( \|H(x_{k-1}, Y_k)\| + \|x_{k-1}\| + \left\|\epsilon_k^{(1)}\right\| \right),$$

$$\text{(By (SKM with Markovian and Additive Noise))}$$

$$\le C_{11} \sum_{k=1}^{n} \alpha_k^2 \left( 2\|x_{k-1}\| + C_H + \left\|\epsilon_k^{(1)}\right\| \right), \qquad \text{(By (12))}$$

$$\le C_{11} \sum_{k=1}^{n} \alpha_k^2 \left( 2C_{10}\tau_{k-1} + 2\sum_{i=1}^{k-1} \alpha_i \left\|\epsilon_i^{(1)}\right\| + C_H + \left\|\epsilon_k^{(1)}\right\| \right). \qquad \text{(Lemma 10)}$$

Because Assumption 4.5 assures us that $\sum_{k=1}^{\infty} \alpha_k \left\| \epsilon_k^{(1)} \right\|$ is finite, then there exists some sample-path dependent constant we denote as $\zeta_{21}$ where,

$$\left\| \sum_{k=1}^{n} \alpha_k \epsilon_k^{(3)} \right\| \leq \zeta_{21} \sum_{k=1}^{n} \alpha_k^2 \left( \tau_{k-1} + \left\| \epsilon_k^{(1)} \right\| \right), \qquad \text{(Assumption 4.5)}$$

$$\leq \zeta_{21} \left( \sum_{k=1}^{n} \alpha_k^2 \tau_k + \sum_{k=1}^{n} \alpha_k^2 \left\| \epsilon_k^{(1)} \right\| \right), \qquad (\tau_k \text{ is increasing})$$

Lemma 8 and Assumption 4.5 then prove the lemma. ∎

**Lemma 22.** *Let $U_n$ be the iterates defined in* (22). *Then if $\sup_n \|U_n\| < \infty$, then we have $U_n \to 0$ almost surely.*

**Proof** We use a stochastic approximation argument to show that $U_n \to 0$. The almost sure convergence of $U_n \to 0$ is given by a generalization of Theorem 2.1 of Borkar (2009), which we present as Theorem 24 in Appendix C for completeness.

We now verify the assumptions of Theorem 24. Beginning with the definition of $\xi_k$ in (21), we have

$$\lim_{n \to \infty} \sup_{j \geq n} \left\| \sum_{k=n}^{j} \alpha_k \xi_k \right\| = \lim_{n \to \infty} \sup_{j \geq n} \left\| \sum_{k=n}^{j} \alpha_k \left( \epsilon_k^{(1)} + \epsilon_k^{(2)} + \epsilon_k^{(3)} \right) \right\|,$$

$$\leq \underbrace{\lim_{n \to \infty} \sup_{j \geq n} \left\| \sum_{k=n}^{j} \alpha_k \epsilon_k^{(1)} \right\|}_{S_1} + \underbrace{\lim_{n \to \infty} \sup_{j \geq n} \left\| \sum_{k=n}^{j} \alpha_k \epsilon_k^{(2)} \right\|}_{S_2} + \underbrace{\lim_{n \to \infty} \sup_{j \geq n} \left\| \sum_{k=n}^{j} \alpha_k \epsilon_k^{(3)} \right\|}_{S_3}.$$

We now bound the three terms in the RHS.

For $S_1$, we have

$$\lim_{n \to \infty} \sup_{j \geq n} \left\| \sum_{k=n}^{j} \alpha_k \epsilon_k^{(1)} \right\| \leq \lim_{n \to \infty} \sup_{j \geq n} \sum_{k=n}^{j} \alpha_k \left\| \epsilon_k^{(1)} \right\| \leq \lim_{n \to \infty} \sum_{k=n}^{\infty} \alpha_k \left\| \epsilon_k^{(1)} \right\| = 0,$$

where we have used the fact that the series $\sum_{k=1}^{n} \alpha_k \left\| \epsilon_k^{(1)} \right\|$ converges by Assumption 4.5 almost surely.

For $S_2$, from (47) in Lemma 20, we have

$$\sum_{k=n}^{j} \alpha_k \epsilon_k^{(2)} = \sum_{k=1}^{j} \alpha_k \epsilon_k^{(2)} - \sum_{k=1}^{n-1} \alpha_k \epsilon_k^{(2)},$$

$$= \alpha_{n-1} \nu(x_n, Y_n) - \alpha_j \nu(x_j, Y_{j+1}) - \sum_{k=n}^{j} (\alpha_{k-1} - \alpha_k) \nu(x_{k-1}, Y_k).$$

Taking the norm and applying the triangle inequality, we have

$$\lim_{n \to \infty} \sup_{j \geq n} \left\| \sum_{k=n}^{j} \alpha_k \epsilon_k^{(2)} \right\| \leq \lim_{n \to \infty} \sup_{j \geq n} \left( \alpha_{n-1} \|\nu(x_n, Y_n)\| + \alpha_j \|\nu(x_j, Y_{j+1})\| \right.$$

$$\left. + \sum_{k=n}^{j} \|(\alpha_{k-1} - \alpha_k) \nu(x_{k-1}, Y_k)\| \right),$$

$$\leq \lim_{n \to \infty} \sup_{j \geq n} \zeta_{12} \left( \alpha_{n-1} \tau_{n-1} + \alpha_j \tau_j + \sum_{k=n}^{\infty} |\alpha_{k-1} - \alpha_k| \tau_{k-1} \right), \quad \text{(Lemma 12)}$$

where the last inequality holds because $\sum_{k=n}^{j}|\alpha_{k-1}-\alpha_k|\tau_{k-1}$ is monotonically increasing. Note that

$$\alpha_n\tau_n = \begin{cases} \mathcal{O}\big(n^{1-2b}\big) & \text{if} \quad \frac{4}{5}<b<1, \\ \mathcal{O}\Big(\frac{\log n}{n}\Big) & \text{if} \quad b=1. \end{cases}$$

Since we have $j\geq n$, then

$$\lim_{n\to\infty}\sup_{j\geq n}\left\|\sum_{k=n}^{j}\alpha_k\epsilon_k^{(2)}\right\| \leq \lim_{n\to\infty}\zeta_{12}\left(2\alpha_{n-1}\tau_{n-1}+\sum_{k=n}^{\infty}|\alpha_{k-1}-\alpha_k|\tau_{k-1}\right)=0$$

where we used the fact that (33) in Lemma 8 and the monotone convergence theorem prove that the series $\sum_{k=1}^{n}|\alpha_k-\alpha_{k+1}|\tau_k$ converges almost surely.

For $S_3$, following the steps in Lemma 21 (which we omit to avoid repetition), we have,

$$\lim_{n\to\infty}\sup_{j\geq n}\left\|\sum_{k=n}^{j}\alpha_k\epsilon_k^{(3)}\right\| \leq \lim_{n\to\infty}\sup_{j\geq n}\zeta_{21}\left(\sum_{k=n}^{j}\alpha_k^2\tau_k+\sum_{k=n}^{j}\alpha_k^2\left\|\epsilon_k^{(1)}\right\|\right).$$

which further implies that

$$\lim_{n\to\infty}\sup_{j\geq n}\left\|\sum_{k=n}^{j}\alpha_k\epsilon_k^{(3)}\right\| \leq \lim_{n\to\infty}\zeta_{21}\left(\sum_{k=n}^{\infty}\alpha_k^2\tau_k+\sum_{k=n}^{\infty}\alpha_k^2\left\|\epsilon_k^{(1)}\right\|\right)=0,$$

where we use the fact that, by (31) in Lemma 8, Assumption 4.5, and the monotone convergence theorem, both series on the RHS series converge almost surely. Therefore we have proven that,

$$\lim_{n\to\infty}\sup_{j\geq n}\left\|\sum_{k=n}^{j}\alpha_k\xi_k\right\|=0 \quad \text{a.s.}$$

thereby verifying Assumption C.1.

Assumption C.2 is satisfied by (13) which is the result of Assumption 4.2. Assumption C.3 is clearly met by the definition of the deterministic learning rates in Assumption 4.4. Demonstrating Assumption C.4 holds, Lemma 14 demonstrates $\{M_n\}$ is square-integrable martingale difference series.

Therefore, by Theorem 24, the iterates $\{U_n\}$ converge almost surely to a possibly sample-path dependent compact connected internally chain transitive set of the following ODE:

$$\frac{dU(t)}{dt}=-U(t). \tag{48}$$

Since the origin is the unique globally asymptotically stable equilibrium point of (48), we have that $U_n\to 0$ almost surely. ∎

**Lemma 23.** *With $\{x_n\}$ defined in (21) and $\{U_n\}$ defined in (22), if $\sum_{k=1}^{\infty}\alpha_k\|U_{k-1}\|$ and $\lim_{n\to\infty}U_n=0$, then $\lim_{n\to\infty}x_n=x_*$ where $x_*\in\mathcal{X}_*$ is a possibly sample-path dependent fixed point.*

**Proof** Following the approach of Bravo & Cominetti (2024), we utilize the estimate for inexact Krasnoselskii-Mann iterations of the form (IKM) presented in Lemma 4 to prove the convergence of (SKM with Markovian and Additive Noise). Using the definition of $\{U_n\}$ in (22), we then let $z_0=x_0$ and define $z_n\doteq x_n-U_n$, which gives

$$\begin{aligned} z_{n+1} &= (1-\alpha_{n+1})x_n+\alpha_{n+1}(h(x_n)+M_{n+1}+\xi_{n+1}) \\ &\quad -((1-\alpha_{n+1})U_n+\alpha_{n+1}(M_{n+1}+\xi_{n+1})) \\ &= (1-\alpha_{n+1})z_n+\alpha_{n+1}h(x_n) \\ &= z_n+\alpha_{n+1}(h(z_n)-z_n+e_{n+1}) \end{aligned}$$

which matches the form of (IKM) with $e_n = h(x_{n-1}) - h(z_{n-1})$. Due to the non-expansivity of $h$ from (13), we have

$$\|e_{n+1}\| = \|h(x_n) - h(z_n)\| \le \|x_n - z_n\| = \|U_{n+1}\|$$

The convergence of $x_n$ then follows directly from Lemma 4 which gives $\lim_{n\to\infty} z_n = x_*$ for some $x_* \in \mathcal{X}_*$, and therefore $\lim_{n\to\infty} x_n = \lim_{n\to\infty} z_n + U_n = x_*$. We note that here $e_n$ is stochastic while the (IKM) result in Lemma 4 considers a deterministic noise. This means here we apply Lemma 4 for each sample path. ∎

## C  EXTENSION OF THEOREM 2.1 OF BORKAR (2009)

In this section, we present a simple extension of Theorem 2 from (Borkar, 2009) for completeness. Readers familiar with stochastic approximation theory should find this extension fairly straightforward. Originally, Chapter 2 of (Borkar, 2009) considers stochastic approximations of the form,

$$y_{n+1} = y_n + \alpha_n(h(y_n) + M_{n+1} + \xi_{n+1}) \tag{49}$$

where it is assumed that $\xi_n \to 0$ almost surely. However, our work requires that we remove the assumption that $\xi_n \to 0$, and replace it with a more mild condition on the asymptotic rate of change of $\xi_n$, akin to Kushner & Yin (2003).

**Assumption C.1.** *For any $T > 0$,*

$$\lim_{n\to\infty} \sup_{n \le j \le m(n,T)} \left\| \sum_{i=n}^{j} \alpha_i \xi_i \right\| = 0 \quad a.s.$$

*where $m(n,T) \doteq \min\left\{ k \mid \sum_{i=n}^{k} \alpha(i) \ge T \right\}$.*

The next four assumptions are the same as the remaining assumptions in Chapter 2 of Borkar (2009).

**Assumption C.2.** *The map $h$ is Lipschitz: $\|h(x) - h(y)\| \le L\|x - y\|$ for some $0 < L < \infty$.*

**Assumption C.3.** *The stepsizes $\{\alpha_n\}$ are positive scalars satisfying*

$$\sum_n \alpha_n = \infty, \sum_n \alpha_n^2 < \infty$$

**Assumption C.4.** *$\{M_n\}$ is a martingale difference sequence w.r.t the increasing family of $\sigma$-algebras*

$$\mathcal{F}_n \doteq \sigma(y_m, M_m, m \le n) = \sigma(y_0, M_1, \ldots, M_n), n \ge 0.$$

*That is,*

$$\mathbb{E}\left[M_{n+1} | \mathcal{F}_n\right] = 0 \quad a.s. \quad , n \ge 0.$$

*Furthermore, $\{M_n\}$ are square-integrable with*

$$\mathbb{E}\left[\|M_{n+1}\|^2 | \mathcal{F}_n\right] \le K\left(1 + \|x_n\|^2\right) \quad a.s. \quad , n \ge 0,$$

*for some constant $K > 0$*

**Assumption C.5.** *The iterates of (49) remain bounded almost surely, i.e.,*

$$\sup_n \|y_n\| < \infty$$

**Theorem 24** (Extension of Theorem 2.1 from Borkar (2009)). *Let Assumptions C.1, C.2, C.3, C.4, C.5 hold. Almost surely, the sequence $\{y_n\}$ generated by (49) converges to a (possibly sample-path dependent) compact connected internally chain transitive set of the ODE*

$$\frac{dy(t)}{dt} = h(y(t)). \tag{50}$$

**Proof** We now demonstrate that even with the relaxed assumption on $\xi_n$, we can still achieve the same almost sure convergence of the iterates achieved by Borkar (2009). Following Chapter 2 of Borkar (2009), we construct a continuous interpolated trajectory $\bar{y}(t), t \geq 0$, and show that it asymptotically approaches the solution set of (50) almost surely. Define time instants $t(0) = 0, t(n) = \sum_{m=0}^{n-1} \alpha_m, n \geq 1$. By assumption C.3, $t(n) \uparrow \infty$. Let $I_n \doteq [t(n), t(n+1)], n \geq 0$. Define a continuous, piece-wise linear $\bar{y}(t), t \geq 0$ by $\bar{y}(t(n)) = y_n, \ n \geq 0$, with linear interpolation on each interval $I_n$:

$$\bar{y}(t) = y_n + (y_{n+1} - y_n)\frac{t - t(n)}{t(n+1) - t(n)}, t \in I_n$$

It is worth noting that $\sup_{t \geq 0}\|\bar{y}(t)\| = \sup_n \|y_n\| < \infty$ almost surely by Assumption C.5. Let $y^s(t), t \geq s$, denote the unique solution to (50) 'starting at s':

$$\frac{dy^s(t)}{dt} = h(y^s(t)), t \geq s,$$

with $y^s(s) = \bar{y}(s), s \in \mathbb{R}$. Similarly, let $y_s(t), t \geq s$, denote the unique solution to (50) 'ending at s':

$$\frac{dy_s(t)}{dt} = h(y_s(t)), t \leq s,$$

with $y_s(s) = \bar{y}(s), s \in \mathbb{R}$. Define also

$$\zeta_n = \sum_{m=0}^{n-1} \alpha_m(M_{m+1} + \xi_{m+1}), \ n \geq 1 \tag{51}$$

**Lemma 25** (Extension of Theorem 1 from Borkar (2009)). *Let $C.1 - C.5$ hold. We have for any $T > 0$,*

$$\lim_{s \to \infty} \sup_{t \in [s, s+T]} \|\bar{y}(t) - y^s(t)\| = 0, \quad \text{a.s.}$$

$$\lim_{s \to \infty} \sup_{t \in [s, s+T]} \|\bar{y}(t) - y_s(t)\| = 0, \quad \text{a.s.}$$

**Proof** Let $t(n+m)$ be in $[t(n), t(n)+T]$. Let $[t] \doteq \max\{t(k) : t(k) \leq t\}$. Then,

$$\bar{y}(t(n+m)) = \bar{y}(t(n)) + \sum_{k=0}^{m-1} \alpha_{n+k}h(\bar{y}(t(n+k))) + \delta_{n,n+m} \quad \text{(2.1.6 in Borkar (2009))} \tag{52}$$

where $\delta_{n,n+m} \doteq \zeta_{n+m} - \zeta_n$. Borkar (2009) then compares this with

$$y^{t(n)}(t(n+m)) = \bar{y}(t(n)) + \sum_{k=0}^{m-1} \alpha_{n+k}h\Big(y^{t(n)}(t(n+k))\Big)$$

$$+ \int_{t(n)}^{t(n+m)} \Big(h\Big(y^{t(n)}(z)\Big) - h\Big(y^{t(n)}([z])\Big)\Big)dz. \quad \text{(2.1.7 in Borkar (2009))}$$

Next, Borkar (2009) bounds the integral on the right-hand side by proving

$$\left\|\int_{t(n)}^{t(n+m)} \Big(h\Big(y^{t(n)}(t)\Big) - h\Big(y^{t(n)}([t])\Big)\Big)dt\right\| \leq C_T L \sum_{k=0}^{\infty} \alpha_{n+k}^2 \xrightarrow{n\uparrow\infty} 0, \quad \text{a.s.} \quad \text{(2.1.8 in Borkar (2009))}$$

where $C_T \doteq \|h(0)\| + L(C_0 + \|h(0)\|T)e^{LT} < \infty$ almost surely and $C_0 \doteq \sup_n \|y_n\| < \infty$ a.s. by Assumption C.5.

Then, we can subtract (2.1.7) from (2.1.6) and take norms, yielding

$$\left\|\bar{y}(t(n+m)) - y^{t(n)}(t(n+m))\right\| \leq L \sum_{i=0}^{m-1} \alpha_{n+i}\left\|\bar{y}(t(n+i)) - y^{t(n)}(t(n+i))\right\|$$

$$+ C_T L \sum_{k \geq 0} \alpha_{n+k}^2 + \sup_{0 \leq k \leq m(n,T)} \|\delta_{n,n+k}\|. \tag{53}$$

The key difference between (53) and the analogous equation in Borkar (2009) Chapter 2, is that we replace the $\sup_{k \geq 0}$ with a $\sup_{0 \leq k \leq m(n,T)}$. The reason we can make this change is that we defined $t(n+m)$ to be in the range $[t(n), t(n) + T]$. Recall that we also defined $m(n,T) \doteq \min \left\{ k \mid \sum_{i=n}^{k} \alpha(i) \geq T \right\}$ in Assumption C.1, so we therefore know that $m \leq m(n,T)$ in (52). Borkar (2009) unnecessarily relaxes this for notation simplicity, but a similar argument can be found in Kushner & Yin (2003).

Also, we have,

$$\|\delta_{n,n+k}\| = \|\zeta_{n+k} - \zeta_n\|,$$

$$= \left\| \sum_{i=n}^{k} \alpha_i (M_{i+1} + \xi_{i+1}) \right\|, \qquad \text{(by (51))}$$

$$\leq \left\| \sum_{i=n}^{k} \alpha_i M_{i+1} \right\| + \left\| \sum_{i=n}^{k} \alpha_i \xi_{i+1} \right\|.$$

Borkar (2009) proves that $\left( \sum_{i=0}^{n-1} \alpha_i M_{i+1}, \mathcal{F}_n \right)$, $n \geq 1$ is a zero mean, square-integrable martingale. By C.3, C.4, C.5,

$$\sum_{n \geq 0} \mathbb{E} \left[ \left\| \sum_{i=0}^{n} \alpha_i M_{i+1} - \sum_{i=0}^{n-1} \alpha_i M_{i+1} \right\| \,\bigg|\, \mathcal{F}_n \right] = \sum_{n \geq 0} \mathbb{E} \left[ \|M_{n+1}\|^2 \,\big|\, \mathcal{F}_n \right] < \infty.$$

Therefore, the martingale convergence theorem gives the almost sure convergence of $\left( \sum_{i=n}^{k} \alpha_i M_{i+1}, \mathcal{F}_n \right)$ as $n \to \infty$. Combining this with assumption C.1 yields,

$$\lim_{n \to \infty} \sup_{0 \leq k \leq m(n,T)} \|\delta_{n,n+k}\| = 0 \quad \text{a.s.}$$

Using the definition of $K_{T,n} \doteq C_T L \sum_{k \geq 0} \alpha_{n+k}^2 + \sup_{0 \leq k \leq m(n,T)} \|\delta_{n,n+k}\|$ given by Borkar (2009), we have proven that our slightly relaxed assumption still yields $K_{T,n} \to 0$ almost surely as $n \to \infty$. The rest of the argument for the proof of the theorem in Borkar (2009) holds without any additional modification. ∎

Having proven Lemma 25, the analysis and proof presented for Theorem 2 in Borkar (2009) applies directly, yielding our desired extended result.

∎

# D RESPONSE TO REVIEWER PPQC

In this section, we address the reviewer PPQC's suggestion that the almost sure convergence of the average-reward TD update can be directly inferred from existing results. The reviewer posits that the convergence of linear TD with a special feature matrix implies the convergence of our tabular TD, potentially rendering our analysis unnecessary. We demonstrate that this argument only holds if expected updates are considered and the reward is always 0 (i.e., $r(s) = 0$).

The outline of the argument proposed by the reviewer is as follows. Let $N \doteq |\mathcal{S}|$. Consider the expected updates of (Average Reward TD) which can be expressed as

$$\bar{v}_{t+1} = \bar{v}_t + \alpha_t (D(P - I)\bar{v}_t + D(r - J_\pi e_N)), \tag{54}$$

where $e_N$ denotes the $N$ dimensional all-one column vector. We can define iterates $\theta_t \in \mathbb{R}^{N-1}$ as

$$\theta_{t+1} = \theta_t + \alpha_t \big( \Phi^\top D(P - I)\Phi \theta_t + \Phi^\top D(r - J_\pi e_N) \big), \tag{55}$$

where $\Phi \in \mathbb{R}^{N \times (N-1)}$ is the feature matrix to be tuned. Let $u_t = \Phi\theta_t$ be the corresponding value function, we then have,

$$u_{t+1} = u_t + \alpha_t \left( \Phi\Phi^\top D(P - I)u_t + \Phi\Phi^\top D(r - J_\pi e_N) \right).$$

The reviewer's claim is that under some smart construction of $\Phi$, two properties can be achieved. First, the matrix $A \doteq \Phi^\top D(P - I)\Phi \in \mathbb{R}^{(N-1) \times (N-1)}$ is negative definite. Second, there exists some $\{c_t \in \mathbb{R}\}$ such that $u_t = \bar{v}_t + c_t e_N$ for all $t$. If both hold, the convergence of $\bar{v}_t$ would be trivial. We, however, believe that this argument only holds if the reward is always 0 (i.e., $r(s) = 0$) and expected updates are considered. For the general stochastic update in (Average Reward TD) with generic reward, this argument does not hold.

### D.1 ANALYSIS OF EXPECTED UPDATES WITH $r = 0$

First we will demonstrate our understanding of the reviewer's point by proving that the reviewer is correct in the case when the reward $r$ is zero, and when we consider the expected updates of (Average Reward TD) as written in (54). Let $k$ be a constant to be tuned. Recall $D \in \mathbb{R}^{N \times N}$ is a diagonal matrix with the diagonal being the stationary distribution $d_\pi$. Following the reviewer's comment, let us define the features $\Phi \in \mathbb{R}^{N \times N-1}$ as

$$\Phi \doteq \begin{bmatrix} I_{N-1} \\ -ke_{N-1}^\top \end{bmatrix}. \tag{56}$$

When $r = 0$, the updates become

$$\theta_{t+1} = \theta_t + \alpha_t \left( \Phi^\top D(P - I)\Phi\theta_t \right),$$
$$u_{t+1} = u_t + \alpha_t \left( \Phi\Phi^\top D(P - I)u_t \right),$$
$$\bar{v}_{t+1} = \bar{v}_t + \alpha_t \left( D(P - I)\bar{v}_t \right).$$

Our goal is to show that,

$$u_t = \bar{v}_t + c_t e_N, \tag{57}$$

for some $c_t \in \mathbb{R}$. To establish this, we define the difference $\delta_t = u_t - \bar{v}_t$ and analyze its evolution,

$$\begin{aligned}
\delta_{t+1} &= u_{t+1} - \bar{v}_{t+1} \\
&= \left( u_t + \alpha_t \Phi\Phi^\top D(P - I)u_t \right) - \left( \bar{v}_t + \alpha_t D(P - I)\bar{v}_t \right) \\
&= \delta_t + \alpha_t \left( \Phi\Phi^\top D(P - I)u_t - D(P - I)\bar{v}_t \right) \\
&= \delta_t + \alpha_t \left( \Phi\Phi^\top D(P - I)(u_t - \bar{v}_t) + \left( \Phi\Phi^\top - I \right) D(P - I)\bar{v}_t \right) \\
&= \delta_t + \alpha_t \left( \Phi\Phi^\top D(P - I)\delta_t + \left( \Phi\Phi^\top - I \right) D(P - I)\bar{v}_t \right).
\end{aligned}$$

We can prove by induction that with a careful choice of $k$, $\delta_t = c_t e$ for all $t$. First, let us define $\delta_0 = 0$. Then, the inductive hypothesis is $\delta_t = c_t e_N$ for some $c_t$. Now we will show that $\delta_{t+1} = c_{t+1}e_N$. It can be shown that $\Phi\Phi^\top D(P - I)\delta_t = 0$ when $\delta_t$ is some scalar multiple of $e_N$. Therefore, the update can be simplified to

$$\delta_{t+1} = \delta_t + \alpha_t \left( \left( \Phi\Phi^\top - I \right) D(P - I)\bar{v}_t \right).$$

Next, we compute $\left( \Phi\Phi^\top - I \right) D(P - I)\bar{v}_t$ and show that it is proportional to $e_N$. Beginning with the definition of $\Phi$ in (56), we have

$$\begin{aligned}
\Phi\Phi^\top &= \begin{bmatrix} I_{N-1} \\ -ke_{N-1}^\top \end{bmatrix} \begin{bmatrix} I_{N-1} & -ke_{N-1} \end{bmatrix} \\
&= \begin{bmatrix} I_{N-1} & -ke_{N-1} \\ -ke_{N-1}^\top & (N-1)k^2 \end{bmatrix}
\end{aligned}$$

Subtracting the identity matrix gives,

$$\Phi\Phi^\top - I = \begin{bmatrix} 0_{(N-1)} & -ke_{N-1} \\ -ke_{N-1}^\top & (N-1)k^2 - 1 \end{bmatrix}$$

where $0_{N-1}$ refers to the $(N-1) \times (N-1)$ dimensional all-zero matrix. Then we have

$$(\Phi\Phi^\top - I)D(P-I)\bar{v}_t = \begin{bmatrix} 0_{(N-1)} & -ke_{N-1} \\ -ke_{N-1}^\top & (N-1)k^2 - 1 \end{bmatrix} D(P-I)\bar{v}_t,$$

$$= \begin{bmatrix} a_t e_{N-1} \\ b_t \end{bmatrix}.$$

where we define $a_t \in \mathbb{R}$ as the first $N-1$ entries of the resulting vector which all share the same value. We use $b_t \in \mathbb{R}$ to denote the $N$-th entry of the resulting column vector. We can see that,

$$a_t \doteq \begin{bmatrix} 0_{1\times(N-1)} & -k \end{bmatrix} D(P-I)\bar{v}_t,$$

and

$$b_t = \begin{bmatrix} -ke_{N-1}^\top & (N-1)k^2 - 1 \end{bmatrix} D(P-I)\bar{v}_t,$$
$$= (\begin{bmatrix} -ke_{N-1}^\top & -k \end{bmatrix} + \begin{bmatrix} 0_{1\times(N-1)} & (N-1)k^2 - 1 + k \end{bmatrix})D(P-I)\bar{v}_t,$$
$$= \begin{bmatrix} 0_{1\times(N-1)} & (N-1)k^2 - 1 + k \end{bmatrix})D(P-I)\bar{v}_t.$$

Therefore if we want $a_t = b_t$ we can solve for $k$,

$$(N-1)k^2 - 1 + k = -k,$$
$$(N-1)k^2 + 2k - 1 = 0,$$
$$k = \frac{-2 + \sqrt{4N}}{2(N-1)} \tag{58}$$

which gives $a_t = b_t$, and $\delta_{t+1} = \delta_t + \alpha_t b_t e$.

However, we do not think the same argument will go through for stochastic updates with a generic reward. In the following sections we will demonstrate that the approach discussed above does not apply to stochastic updates with general rewards. Now we show that even if it applied, there is still problem. To see this, we telescope the recursion of $\delta_t$ and obtain

$$u_t = \bar{v}_t + \left(\sum_{i=0}^{t-1} \alpha_i b_i\right) e,$$

$$\bar{v}_t = u_t - c_t e,$$

with

$$c_t = \sum_{i=0}^{t-1} \alpha_i b_i.$$

We recall that

$$b_i = \begin{bmatrix} 0_{1\times(N-1)} & -k \end{bmatrix} D(P-I)\bar{v}_i$$
$$= \begin{bmatrix} 0_{1\times(N-1)} & -k \end{bmatrix} D(P-I)(u_i - \delta_i)$$
$$= \begin{bmatrix} 0_{1\times(N-1)} & -k \end{bmatrix} D(P-I)u_i.$$

We know that $\{u_t\}$ converges 0. So $\{b_i\}$ converges to 0. But **this does not mean $\{c_t\}$ converges.** To establish the convergence of $\{c_t\}$, we have to know the almost sure convergence rate of $u_t$. With the expected updates, this is not hard. But with stochastic updates, to our knowledge, there is no existing result showing the almost sure convergence rate of average reward linear TD. **If we cannot show $\{c_t\}$ converges, then the reviewer's approach cannot prove that $\{\bar{v}_t\}$ converge to a single fixed point.** It can at most say $\{\bar{v}_t\}$ converge to a (possibly unbounded) set of fixed points.

## D.2 ANALYSIS OF EXPECTED UPDATES WITH UNKNOWN $J_\pi$

When we remove the assumption that $r$ is zero and use $J_t$ generated by (Average Reward TD) instead of $J_\pi$, the equivalence cannot be proven. In this case the updates can be written as

$$\theta_{t+1} = \theta_t + \alpha_t\big(\Phi^\top D(P-I)\Phi\theta_t + \Phi^\top D(r - J_t e_N)\big),$$
$$u_{t+1} = u_t + \alpha_t\big(\Phi\Phi^\top D(P-I)u_t + \Phi\Phi^\top D(r - J_t e_N)\big).$$

Once again, the goal is to show that if we construct $\Phi$ as,

$$\Phi \doteq \begin{bmatrix} I_{N-1} \\ -ke_{N-1}^\top \end{bmatrix},$$

then we can prove that $u_t = \bar{v}_t + c_t e_N$. To this end, we once again define the difference $\delta_t = u_t - \bar{v}_t$, with the goal of showing that $\delta_t$ is proportional to $e$.

$$\begin{aligned}
\delta_{t+1} &= u_{t+1} - \bar{v}_{t+1} \\
&= \left(u_t + \alpha_t \Phi\Phi^\top D(P-I)u_t + \alpha_t \Phi\Phi^\top D(r - J_t e_N)\right) \\
&\quad - \left(\bar{v}_t + \alpha_t D(P-I)\bar{v}_t + \alpha_t D(r - J_t e_N)\right) \\
&= \delta_t + \alpha_t \left(\Phi\Phi^\top D(P-I)u_t - D(P-I)\bar{v}_t\right) + \alpha_t \left(\Phi\Phi^\top D(r - J_t e_N) - D(r - J_t e_N)\right) \\
&= \delta_t + \alpha_t \left(\Phi\Phi^\top D(P-I)\delta_t + \left(\Phi\Phi^\top - I\right) D(P-I)\bar{v}_t\right) + \alpha_t \left(\left(\Phi\Phi^\top - I\right) D(r - J_t e_N)\right).
\end{aligned}$$

Once again, we have $\Phi\Phi^\top D(P-I)\delta_t = 0$ when $\delta_t$ is proportional to $e$, so

$$\delta_{t+1} = \delta_t + \alpha_t \left( \underbrace{\left(\Phi\Phi^\top - I\right) D(P-I)\bar{v}_t}_{S_1} + \underbrace{\left(\Phi\Phi^\top - I\right) D(r - J_t e_N)}_{S_2} \right).$$

Previously, we showed that if we choose $k = \frac{-2+\sqrt{4N}}{2(N-1)}$, $S_1$ can be written as $b_t e$ for some scalar $b_t$. However, given that $r$ is now non-zero and $J_t$ can literally be any number along the sample path, we cannot prove that $S_2$ is also proportional to $e$, which is required to satisfy (57). To see this, we have,

$$\begin{aligned}
(\Phi\Phi^\top - I)D(r - J_t e_N) &= \begin{bmatrix} 0_{N-1} & -ke_{N-1} \\ -ke_{N-1}^\top & (N-1)k^2 - 1 \end{bmatrix} D(r - J_t e_N), \\
&= \begin{bmatrix} f_t e_{N-1} \\ g_t \end{bmatrix}.
\end{aligned}$$

where we define $f_t \in \mathbb{R}$ as the first $N-1$ entries of the resulting vector which all share the same value. We use $g_t \in \mathbb{R}$ to denote the $N$-th entry of the resulting column vector. We have,

$$f_t \doteq \begin{bmatrix} 0_{1\times(N-1)} & -k \end{bmatrix} D(r - J_t e_N),$$

and

$$\begin{aligned}
g_t &= \begin{bmatrix} -ke_{N-1}^\top & (N-1)k^2 - 1 \end{bmatrix} D(r - J_t e_N), \\
&= \left(\begin{bmatrix} -ke_{N-1}^\top & -k \end{bmatrix} + \begin{bmatrix} 0_{1\times(N-1)} & (N-1)k^2 - 1 + k \end{bmatrix}\right) D(r - J_t e_N), \\
&= -k(J_\pi - NJ_t) + \begin{bmatrix} 0_{1\times(N-1)} & (N-1)k^2 - 1 + k \end{bmatrix} D(r - J_t e_N) \\
&= -k(J_\pi - NJ_t) + \begin{bmatrix} 0_{1\times(N-1)} & -k \end{bmatrix} D(r - J_t e_N) \\
&= -k(J_\pi - NJ_t) + f_t,
\end{aligned}$$

where we recall $k$ is defined in (58). **Since $J_t$ can be an arbitrary number, there is no way that $g_t = f_t$ holds for all** $t$. We believe the fundamental cause is that $e_N^\top D(P-I) = 0$ but $e_N^\top D(r - J_t e_N)$ is arbitrary. Even if $J_t = J_\pi$, we still have $e_N^\top D(r - J_\pi e_N) \neq 0$. To make $e_N^\top D(r - J_\pi e_N) = 0$, we have to artificially multiply $r$ by $N$ in (Average Reward TD). But even with this, if $J_t$ is used, it still does not work. This demonstrates the complexity of the problem when stochastic updates are involved. We recall now only $J_t$ is stochastic. In the next section, we show the problem is harder if we consider the full stochastic setting.

### D.3   ANALYSIS OF STOCHASTIC UPDATES

For simplicity, we will consider the case where $J_\pi$ is known and does not need to be estimated. Let $x(s) \in \mathbb{R}^N$ denote the one-hot vector where only the $s$-th element is 1. Use shorthand $x_t \doteq x(S_t)$ and $r_t \doteq r(S_t)$. The (Average Reward TD) is then

$$v_{t+1} = v_t + \alpha_t(x_t(x_{t+1}^\top - x_t^\top)v_t + x_t(r_t - J_\pi))$$

Let $\phi(s) \in \mathbb{R}^{N-1}$ denote the $s$-th row of $\Phi$, i.e., $\phi(s)$ is the feature of $s$. We will use $\phi_t \in \mathbb{R}^{N-1}$ as shorthand to denote the feature $\phi(S_t)$ which is the row of $\Phi$ corresponding to the state $S_t$. Then this gives the updates

$$\theta_{t+1} = \theta_t + \alpha_t\big(\phi_t(\phi_{t+1}^\top - \phi_t^\top)\theta_t + \phi_t(r_t - J_\pi)\big).$$

We have $u_t \doteq \Phi\theta_t$, which gives,

$$u_{t+1} = u_t + \alpha_t\big(\Phi\phi_t(\phi_{t+1}^\top - \phi_t^\top)\theta_t + \Phi\phi_t(r_t - J_\pi)\big),$$
$$u_{t+1} = u_t + \alpha_t(\Phi\phi_t(u_t(S_{t+1}) - u_t(S_t)) + \Phi\phi_t(r_t - J_\pi)),$$
$$= u_t + \alpha_t(\Phi\phi_t(x_{t+1}^\top - x_t^\top)u_t + \Phi\phi_t(r_t - J_\pi))$$

Once again, the goal is to show that if we construct $\Phi$ as,

$$\Phi \doteq \begin{bmatrix} I_{N-1} \\ -ke_{N-1}^\top \end{bmatrix},$$

then we can prove that $u_t = v_t + c_t e_N$. To this end, we once again define the difference $\delta_t = u_t - v_t$, with the goal of showing that $\delta_t$ is proportional to $e$.

$$\delta_{t+1} = u_{t+1} - v_{t+1}$$
$$= \big(u_t + \alpha_t(\Phi\phi_t(x_{t+1}^\top - x_t^\top)u_t + \Phi\phi_t(r_t - J_\pi))\big)$$
$$- \big(v_t + \alpha_t(x_t(x_{t+1}^\top - x_t^\top)v_t + x_t(r_t - J_\pi))\big)$$
$$= \delta_t + \alpha_t\big(\Phi\phi_t(x_{t+1}^\top - x_t^\top)u_t - x_t(x_{t+1}^\top - x_t^\top)v_t\big) + \alpha_t(\Phi\phi_t(r_t - J_\pi) - x_t(r_t - J_\pi))$$
$$= \delta_t + \alpha_t\left(\underbrace{\Phi\phi_t(x_{t+1}^\top - x_t^\top)u_t}_{S_1} - \underbrace{x_t(x_{t+1}^\top - x_t^\top)v_t}_{S_2} + \underbrace{(\Phi\phi_t - x_t)(r_t - J_\pi)}_{S_3}\right)$$

Let $S_t$ be one of the first $N-1$ states. Without loss of generality, let the features of the current state $S_t$ correspond to the first row of $\Phi$. Under this construction of $\Phi$ from (56), we have

$$(\Phi\phi_t) = \begin{bmatrix} 1 \\ 0_{(N-2)\times 1} \\ -k \end{bmatrix}.$$

Therefore, regardless of the current composition of $u_t$, the term $S_1$ can only take the form of

$$S_1 = \begin{bmatrix} a_t \\ 0_{(N-2)\times 1} \\ -ka_t \end{bmatrix},$$

where $a_t \doteq (x_{t+1}^\top - x_t^\top)u_t \in \mathbb{R}$. Now if we consider $S_2$, it can only take the form of,

$$S_2 = \begin{bmatrix} b_t \\ 0_{(N-2)\times 1} \\ 0 \end{bmatrix},$$

where $b_t = (x_{t+1}^\top - x_t^\top)v_t$. Finally for the form of $S_3$, we first note that

$$(\Phi\phi_t - x_t) = \begin{bmatrix} 0 \\ 0_{((N-2)\times 1)} \\ -k \end{bmatrix},$$

which implies $S_3$ takes the form of

$$S_3 = \begin{bmatrix} 0 \\ 0_{((N-2)\times 1)} \\ -kd_t \end{bmatrix},$$

where $d_t = (r_t - J_\pi) \in \mathbb{R}$. Then we have,

$$\delta_{t+1} = \delta_t + \alpha_t \left( \underbrace{\begin{bmatrix} a_t \\ 0_{(N-2)\times 1} \\ -ka_t \end{bmatrix}}_{S_1} - \underbrace{\begin{bmatrix} b_t \\ 0_{(N-2)\times 1} \\ 0 \end{bmatrix}}_{S_2} + \underbrace{\begin{bmatrix} 0 \\ 0_{((N-2)\times 1)} \\ -kd_t \end{bmatrix}}_{S_3} \right)$$

$$= \delta_t + \alpha_t \left( \begin{bmatrix} a_t - b_t \\ 0_{((N-2)\times 1)} \\ -k(a_t + d_t) \end{bmatrix} \right)$$

In order for $\delta_{t+1}$ to be proportional to $e$, it is therefore necessary that $S_1 - S_2 + S_3 = 0$ since we can see that $S_1$, $S_2$, and $S_3$ all have 0 in the middle $N - 2$ entries which are completely independent from any choice of $k$. Since $r_t$ and $x_{t+1}$ depend on the specific realization of the random trajectory $S_t, S_{t+1}$, we cannot say that $a_t = -d_t$ for all $t$. Here if we replace $J_\pi$ with $J_t$, it becomes even more problematic. $r_t$ is at least somehow related to $J_\pi$ but $J_t$ can literally be anything in a sample path. **Therefore, in order for $\delta_t = c_t e$ for all $t$, it must be the case that $k = 0$, which contradicts the requirement that $A$ be Hurwitz**.

In conclusion, although the reviewer is correct that the almost-sure convergence of $\theta_t$ in (55) directly implies the almost sure convergence of the expected iterates (54) of (Average Reward TD) in the special case when $r = 0$, this statement does not hold when we consider non-zero reward, as well as the actual stochastic update (Average Reward TD).

