# OpenReview forum: "Almost Sure Convergence of Average Reward Temporal Difference Learning"
_ICLR.cc/2025/Conference — Submitted to ICLR 2025_

### Official Review · Reviewer_Mp5r · 2024-10-29

**Soundness:** 4
**Presentation:** 3
**Contribution:** 2
**Rating:** 3
**Confidence:** 3

**Summary:**

The paper provides an analysis of tabular average reward TD under the additive noise assumption and assuming a Markov chain. The analysis extends the result of SKM to Markovian noise case.

**Strengths:**

1. The paper is well-written and provide clear comparison with related works.


2. The authors almost sure convergence of tabular average reward TD under some restricted assumptions. The work may provide some hints on further proving its convergence under milder assumptions.

**Weaknesses:**

1. My major concern is that as the authors mentioned, the additive noise assumption in Assumption 4.5 is quite strong assumption. For example, in bounding, $\bar{\bar{\epsilon}}^{(1)}_n$, the additive noise assumption plays a key role. At least in the context of RL, this assumption seems to be not a mild assumption, and not commonly use used, as opposed to the argument in the abstract by the authors. With this assumption, I believe many open problems in RL can be solved.


2. The analysis of stochastic approximation with Markovian noise has been well-studied in the literature, in particular using the Poisson equation. Therefore, it is questionable, what is the difficulty of applying such techinque to the anlaysis of SKM.

**Questions:**

1. $e$ in equation (1) has not been defined previously.

---

> ### Author Response · Authors · 2024-11-27
> **Response to Reviewer Mp5r**
>
> >Weaknesses: 1. My major concern is that as the authors mentioned, the additive noise assumption in Assumption 4.5 is quite strong assumption. For example, in bounding, the additive noise assumption plays a key role. At least in the context of RL, this assumption seems to be not a mild assumption, and not commonly use used, as opposed to the argument in the abstract by the authors. With this assumption, I believe many open problems in RL can be solved.
>
> To clarify, while Assumption 4.5 concerning $\epsilon_1$ might appear to be a strong assumption in isolation, we already allow for **Markovian noise** $Y_t$ in our analysis, without imposing any assumptions on its structure. This represents a significant advancement, as existing works do not address the case of non-expansive mappings with Markovian noise. The additive noise $\epsilon_1$ is introduced only to establish a specific proof for average reward TD, and in many practical applications, $\epsilon_1 = 0$ while still accommodating the much less restrictive Markovian noise, which is weaker than i.i.d. or Martingale noise. Therefore, the assumption does not limit the applicability of our framework.
>
> Furthermore, Theorem 2, which applies the stochastic approximation result to average reward TD, does not rely on any assumptions regarding $\epsilon_1$. Instead, it builds solely on Assumption 5.1 concerning the egodicity of the Markov Chain, which together with the update rule for average reward TD ensures the conditions of Assumptions 4.1–4.5 are met.
>
>
> > 2. The analysis of stochastic approximation with Markovian noise has been well-studied in the literature, in particular using the Poisson equation. Therefore, it is questionable, what is the difficulty of applying such technique to the analysis of SKM.
>
> We argue that stochastic approximation with Markovian noise for **non-expansive mappings** has received limited attention in the literature. Most works on stochastic approximation, including those by Kushner and Yin or Borkar, do not address this specific setting. If the reviewer is aware of existing works that study stochastic approximation with **non-expansive mappings** and Markovian noise, we would welcome specific references.
>
> Additionally, while stochastic approximation techniques using the Poisson decomposition are well-established, our analysis required bounding the novel terms $\bar{\epsilon_n}^(j)$ and $\bar{\bar{\epsilon_n}}^(j)$ for $j = \{1, 2, 3\}$ to ensure the almost sure stability of our iterates. This task was highly non-trivial and represents a critical component of our proof. We elaborate on the novelty of our approach and its relationship to prior work utilizing Poisson decomposition in stochastic approximation in the second paragraph of the Related Work section.
>
> Finally, while the technique itself may appear straightforward to some, the novelty and utility of our result are independent of the perceived difficulty of the analysis. Importantly, this result enables us to provide the first proof of almost sure convergence for the most prominent algorithm in average reward MDPs after 25 years.
>
> Questions:
>
> >$e$ in equation (1) has not been defined previously.
>
> We explicitly defined $e$ in lines 51-52 as the all-one column vector.

---

> > ### Comment · Reviewer_Mp5r · 2024-11-30
> >
> > Thank you for the detailed comments. While most of my concerns have been addressed, I still have a few remaining points:
> >
> > Although the authors claim that combining non-expansive mappings and Markovian noise has not been studied, and I acknowledge that the result itself can be considered a contribution, I am not fully convinced of its originality or significance. This is because both topics have been well-studied in SKM [Bravo et al., 2024] and in literature of stochastic approximation with Markovian noise.
> >
> >
> > Moreover, the authors claim that there are new terms to bound (in paragraph 2 of related works), which authors claim to be novel and challenging. But I think the claim is insufficient because the terms not appearing in the previous literature does not always mean that it is novel or challenging. There should be more supporting claims or explanation why the terms are novel and they are difficult to bound.
> >
> >
> > Lastly, following the discussion with Reviewer PPqC, I have some concerns regarding the choice of feature matrix. While I do not have a specific suggestion nor the literature seems to have one, it is questionable whether we can definitively conclude that the result of Tsitsiklis and Van Roy does not imply the convergence in a tabular setup. Are there any chance that a nice choice of feature matrix other than Reviewer PPqC suggested, can imply the convergence in the tabular setup?

---

> > > ### Author Response · Authors · 2024-12-02
> > > **Follow-Up Response to Reviewer Mp5r**
> > >
> > > First, we want to thank the reviewer for their engagement during this discussion period. We hope to address the remaining concerns and questions in this follow-up.
> > >
> > > >Although the authors claim that combining non-expansive mappings and Markovian noise has not been studied, and I acknowledge that the result itself can be considered a contribution, I am not fully convinced of its originality or significance. This is because both topics have been well-studied in SKM [Bravo et al., 2024] and in literature of stochastic approximation with Markovian noise
> > >
> > > We appreciate the acknowledgment that our result itself can be considered a contribution. However, we respectfully disagree with the claim that our extension lacks originality or significance. While it is true that stochastic approximation (SA) with Markovian noise and SA with non-expansive mappings have been well-studied **separately**, our work is the first to analyze their intersection and address the challenges that arise in combining them. We welcome any **specific** references or arguments indicating that this combination has been previously studied or that our results follow trivially from existing work.
> > >
> > > Additionally, Bravo et al. (2024) focuses on SA with Martingale difference noise, which crucially assumes that $Y_i$ (in our case) is i.i.d. . This assumption significantly limits its applicability to RL where $Y_i$ is typically Markovian. Our work is significant because it relaxes this restrictive i.i.d. assumption, broadening the set of tools available for studying average-reward RL algorithms.
> > >
> > > >Moreover, the authors claim that there are new terms to bound (in paragraph 2 of related works), which authors claim to be novel and challenging. But I think the claim is insufficient because the terms not appearing in the previous literature does not always mean that it is novel or challenging. There should be more supporting claims or explanation why the terms are novel and they are difficult to bound.
> > >
> > > While the absence of terms in previous literature alone does not guarantee their difficulty, addressing previously unexplored terms or dynamics is a key aspect of theoretical research. Furthermore, we direct the reviewer to Lemmas 17, 18, 20, and 21, where these new terms are bounded. We welcome any **specific** references where these terms have been bounded previously.
> > >
> > > Even if the reviewer perceives the bounding these terms as not particularly challenging, their inclusion enables a substantial extension of SA with non-expansive mappings under Markovian noise and the first proof of almost sure convergence of the fundamental average reward TD algorithm in the tabular setting. We believe that the technical contributions and impact of this work should stand independently of subjective judgements of perceived difficulty.
> > >
> > > >Lastly, following the discussion with Reviewer PPqC, I have some concerns regarding the choice of feature matrix. While I do not have a specific suggestion nor the literature seems to have one, it is questionable whether we can definitively conclude that the result of Tsitsiklis and Van Roy does not imply the convergence in a tabular setup. Are there any chance that a nice choice of feature matrix other than Reviewer PPqC suggested, can imply the convergence in the tabular setup?
> > >
> > > First, we want to emphasize that the argument proposed by Reviewer PPqC (not just the specific feature construction they provide) is incorrect. First, we explicitly prove their construction is incorrect (see Appendix D). Additionally, in our discussion, we provide an additional general argument explaining why *no choice of feature matrix* results in Tsitsilkis and Van Roy automatically implying the almost sure convergence in the tabular setting. We summarize this argument here:
> > >
> > > Assume, hypothetically, that it were somehow proved $v_t = u_t + c_t e$ for general stochastic updates with general rewards (recall that this is not actually proved). While $\theta_t$ converges almost surely (a.s.), implying $u_t$ converges a.s. to a fixed point, our analysis (see Appendix D.1) shows $c_t$ has the form $c_t = \sum_{i=0}^{t-1} \alpha_i z^\top u_i$ for some fixed vector $z$. Crucially, the convergence of $u_i $ does not imply the convergence of $c_t$. Proving convergence of $c_t$ would at least require an a.s. convergence rate for $u_t$, which, to our knowledge, does not exist for average reward linear TD. If we cannot prove $c_t$ converges to a sample path dependent fixed scalar, we cannot prove that $v_t$ converge to a sample path dependent fixed point following the reviewer's approach
> > >
> > > Our method, on the other hand, does prove that $v_t$ almost surely converges to a sample-path-dependent fixed point. We have revised the submission to further clarify this argument and provide additional perspective to demonstrate why the reviewer's proposed approach is insufficient for convergence in the tabular setting.

---

> > > > ### Comment · Reviewer_Mp5r · 2024-12-03
> > > >
> > > > Thanks for the detailed response.
> > > >
> > > > I understand the importance for broadening the tools for studying the average-reward RL algorithms. But my point is that the explanation on the technical novelty and challenges authors claiming are insufficient. If the Lemmas 17-21 are the key ideas that overcome the challenges, then detailed explanation about them will be helpful to understand the novelty and challenges.
> > > >
> > > > Furthermore, I understand that the author's mathematical arguments prove that the case for $r\neq 0$. But can we just use some kind of coordinate change, e.g., $v_t\to v_t-v^*$ for some $v^*\in \mathcal{V}$ and reduce the problem to the case when $r=0$? It is somewhat counter-intuitive that the Reviewer PPqC's argument only holds for $r=0$. Can the authors provide an intuitive reason?

---

> > > > > ### Author Response · Authors · 2024-12-03
> > > > >
> > > > > Thanks for the quick response.
> > > > >
> > > > > > But my point is that the explanation on the technical novelty and challenges authors claiming are insufficient. If the Lemmas 17-21 are the key ideas that overcome the challenges, then detailed explanation about them will be helpful to understand the novelty and challenges.
> > > > >
> > > > > We agree with this and will revise the submission accordingly in next revision.
> > > > >
> > > > > > Can the authors provide an intuitive reason?
> > > > >
> > > > > Yes. We first want to clarify that the Reviewer PPqC's argument only holds when $r=0$ **and** expected updates are used. If $r=0$ but stochastic updates are used, it does not work. Fundamentally, that construction works because the identity $e^\top D(P - I)$. But with stochastic updates, we will not have $D(P - I)$ so the construction will not work. We understand that the coordinate change is a common (and usually successful) way to work with non-zero constant term. However, it will not apply here. Intuitively, this is because as long as $r\neq 0$, we cannot use $J_\pi = 0$ and have to use the stochastic estimate $J_t$, which can literally be any real number along the sample path. As can be seen in our proof, the reviewer PPqC's approach is essentially to solve a system of linear equations to find the smart feature. However, if that system of linear equations involve $J_t$, it is unlikely that a solution can simultaneously hold for all $J_t$ because $J_t$ can be anything. In short, having $r\neq 0$ not only introduces a shift in the coordinate but also introduces another stochasticity ($J_t$).

---

### Official Review · Reviewer_PPQC · 2024-10-30

**Soundness:** 2
**Presentation:** 1
**Contribution:** 1
**Rating:** 1
**Confidence:** 5

**Summary:**

The present paper studies tabular average rewerd temporal difference and proves it converges almost surely to a sample path dependent fixed point.

**Strengths:**

The proofs in this paper seem sound, even if the tools used are much more complicated than the problem.

**Weaknesses:**

In my view, the contributions of this paper are limited, as the problem it addresses is significantly simpler than the authors suggest. At least one-third of the paper is spent attempting to persuade the reader that the problem is more complex than it actually is, taking up considerable space and reducing readability.

Here is a list of concerns:
1. The sections "Hardness in Stability" and "Hardness in Convergence" seem somewhat artificial: (5) and (6) admit explicit solutions, using matrix exponentials and the superposition principle. This approach would make the analysis clearer and directly yield the desired stability property, which is straightforward. It follows simply from the fact that $D(P-I_d)$ admits $A=$ { constant vectors } as its kernel and is Hurwitz on any space complementary to $A$. Similarly, the convergence property mentioned on lines 147–148 is evident using the explicit solution.
2. I disagree with the interpretations presented in the "Hardness with Linear Function Approximation" section. In fact, the results of this paper can be easily derived using standard results on linear function approximation. It suffices to take $K=|S|-1$, $\Phi(i,i)=1-1/|S|$ and $\Phi(j,i)=-1/|S|$ for $0\leq i\leq |S|-1$ and $j\neq i$. This leads to a new algorithm similar to the one studied in this paper, up to an additive constant vector (constant with respect to the state but not the iterative variable). This approach enables straightforward convergence using the ODE method ($L^2$ and almost surely) and provides a much more precise description of the path-dependent limit in Theorem 5.1.
3. Several primary definitions are missing, such as sample path-dependent convergence. Some standard definitions are misused: for instance, (3) is not an Euler discretization of (5), nor is the equation on line 188 an Euler discretization of (9).
4. The main argument in this paper relies on the ODE method, yet I am surprised it is never explicitly cited. Moreover, this method only provides asymptotic convergence results, while the trend is increasingly toward non-asymptotic results. This leads me to think that a thorough discussion of such methods is missing, at least in the "Related Work" section.

**Questions:**

See the weaknesses.

---

> ### Author Response · Authors · 2024-11-12
>
> Thanks for your comments. Before we proceed to a response, could you clarify what $\Phi(i, i)$ is in your construction? It seems sth is missing now.

---

> ### Author Response · Authors · 2024-11-12
>
> We thank the reviewer for clarification. We want to confirm with the reviewer if our understanding of the implication of this construction is correct so we can provide an effective response.
>
> For this new feature matrix, it has the property that $e^\top \Phi w = 0$ for any $w$ where $e$ is an all one vector. Let $v(i) = \Phi(i, :) w$ be the state value approximation for $i$ using $w$ under this feature matrix. Then this construction ensures that $\sum_i v(i) = 0$. Let's assume for now the true value function satisfies $\sum_i v_\pi(i) = 0$. Then by running linear TD with this feature, we are able to converge to the true value function. In other words, linear TD with this feature matrix is a practical algorithm to find the true value function and can achieve the same goal as the tabular TD we studied. Does the reviewer mean that given this linear TD algorithm, there is no need to use the tabular TD we studied at all, or the convergence of this linear TD implies the convergence our tabular TD? If it is the latter case, we would appreciate if the reviewer could give more hints on how the equivalence can be established. In particular, we failed to understand what "constant with respect to the state" mean. Does it mean $\Phi$ is obtained by shifting ("additive constant") our feature $I$ by $1/|S|$? But the last column is dropped after this shifting so we failed to see the equivalence.
>
> We also note that for average reward MDP, we only have $\sum_i d(i) v_\pi(i) = 0$ where $d$ is the stationary distribution and $\sum_i v_\pi(i) = 0$ is not necessarily true. Let $D$ be a diagonal matrix with the diagonal being $d$. Following the reviewer's idea, we might need to consider a feature matrix $\Phi' = D^{-1} \Phi$ so that we have $d^\top \Phi' w = 0$. But the linear TD with $\Phi'$ is not a practical algorithm because $D$ is unknown. Furthermore, it is again not clear how the convergence of linear TD with $\Phi'$ would imply the convergence of our tabular TD and we would appreciate more hints. Now it is not shifting. Instead, it is (1) shifting (2) dropping the last column and (3) rescaling. So we failed to see the equivalence.

---

> ### Comment · Reviewer_PPQC · 2024-11-13
>
> Before addressing the authors' questions, I would like to clarify my previous comment by providing additional details.
>
> First, recall that equation (1) admits a unique solution up to an additive constant, meaning that the set in (2) is one-dimensional. Consequently, finding a solution to (1) is an underdetermined problem. However, by adding a constraint to fix the additive constant, the problem becomes well-posed. The standard condition used in the optimal control community (where this problem is often called "ergodic") is to fix $\sum_{i}v_{\pi}(i)=0$.
>
> To ensure this condition holds, we can adjust the algorithm in (Average Reward TD) as follows:
> $$
> J_{t+1}=J_t+\beta_{t+1}(R_{t+1}-J_t),
> $$
> $$
> u_{t+1}(S_t) =u_t(S_t) + \alpha_{t+1}(1-1/|S|) \delta_{t+1},
> $$
> $$
> u_{t+1}(s) = u_t(s) - \alpha_{t+1}\delta_{t+1}/|S| \text{ for } s\neq S_t,
> $$
> where $\delta_{t+1} = (R_{t+1}-J_t+u_t(S_{t+1})-u_t(S_t))$. My point is that $u$ matches the original process $v$ proposed by the authors in (Average Reward TD) up to an additive constant, i.e., $v_t = u_t + c_te$ where $c_t$ is a real number. The values of $c_t$ can be computed as
> $$
> c_{t+1} = c_t + \alpha_{t+1}TD_{t+1}/|S|.
> $$
> Therefore, one can easily pass from $u$ to $v$ and vice versa.
>
> In my previous comment, I made two key points:
> 1. On the one hand, $u$ is defined by a linear stochastic algorithm, as described in the section "Hardness with linear function approximation," and it satisfies the conditions under which Tsitsiklis & Roy (1999) proved almost sure convergence. Therefore, $u$ converges almost surely to some $u_{\pi}$. Furthermore, it is straightforward to verify that $u_{\pi}$ is a solution to (1).
> 2. On the other hand, the authors claim that $v$ converges to a sample-path-dependent fixed point. I believe they do not adequately define what is meant by a "sample-path-dependent fixed point". I contend that a more precise statement is that $v$ converges up to an additive constant to $u_{\pi}$, with this additive constant, namely $c_t$ as defined above, not converging to any deterministic limit. In other words, the sample-path dependence described by the authors is fully characterized by $u_{\pi}$ and $(c_{t}(\omega))_{t\geq0}$ for almost all $\omega \in \Omega$ (where $\Omega$ represents the ambient probability space).
>
> To summarize, I assert that this perspective simplifies both the problem and the proofs, providing a more precise characterization of the asymptotic behavior of $v$. This is why I believe the contributions of this paper are insufficient for publication in ICLR.
>
> Now, I will briefly respond to the authors' questions:
> 1. Q:' Does the reviewer mean that given this linear TD algorithm, there is no need to use the tabular TD we studied at all, or the convergence of this linear TD implies the convergence our tabular TD?' R: I assert that the two are equivalent since $v_t = u_t + c_te$, implying that studying the asymptotic behavior of one provides the asymptotic behavior of the other.
> 2. Q: 'Does it mean $\Phi$ is obtained by shifting ("additive constant") our feature $I$ by $1/|S|$?' R: The algorithm I propose involves a simple projection step onto the set of vectors $u$ satisfying $\sum_{i}u(i)=0$ after each step of (Average Reward TD). Thus, dim(Im$(\Phi))=|S|-1$, and $\Phi$ should be defined as I proposed (while the precise form of $\Phi$ may differ, it should have the same general structure; I leave the details to the authors).
> 3. Regarding the last paragraph in the latter authors' response :  I agree that $v_{\pi}$ (defined in line 65) does not satisfy $\sum_iv_{\pi}(i)=0$ in general. Nonetheless, we have that $v_{\pi}=u_{\pi}+c_{\pi}e$ for some constant $c_{\pi}$. I acknowledge that I did not address the determination of the constant $c_{\pi}$, nor did the authors with their method. Since both algorithms rely solely on the Bellman equation (1), neither can capture $c_{\pi}$. However, I do not consider this an issue, as knowledge of $v_{\pi}$ up to an additive constant is sufficient and well-established in the optimal control community.

---

> > ### Author Response · Authors · 2024-11-13
> >
> > We thanks the reviewer for the detailed reply but we still failed to get why the proposed $u_t$ will converge. To simplify notations, let's say the reward is always 0.
> > If we want to invoke Tsitsiklis & Roy (1999), we need to compute the expected updates of $u_t$. First, the expectation of $\delta_{t+1}$ given $u_t$ would be $e^\top D(P - I) u_t$, where $e$ is an all one vector. Then the expected updates of $u_t$ can be expressed as
> > $$u_{t+1} = u_t + \alpha_{t+1} (D(P - I) u_t - (e^\top D(P - I) u_t)e / |S| )$$
> > If we multiply by $e^\top$ on both sides of the above update, it can be seen that $e^\top u_{t+1} = e^\top u_t$.
> > Given this update, we would need to study the matrix $A = (I - e e^\top /|S|) D(P - I)$. But actually, we have $e^\top D(P - I) = 0$. So this $A$ is really just $D(P - I)$, which is the same as the expected updates of our tabular TD. As a result, the results in Tsitsiklis & Roy (1999) do not apply. We would appreciate if the reviewer can point out if there is any mistake in our derivation and give more hints on the convergence of $u_t$.
> >
> > We also want to clarify that our Theorem 2 says that for each sample path $\omega$, there exists some fixed point $v(\omega) = v_\pi + c(\omega) e$ such that $v_t(\omega) \to v(\omega)$. Notably, this $c(\omega)$ does not depend on $t$. This is what we mean by sample path dependent fixed point -- it only depends on the sample path but not $t$. We will make more explicit in revision. In the reviewer's result, it seems to have a time dependent $c_t(\omega)$.

---

> > > ### Comment · Reviewer_PPQC · 2024-11-13
> > >
> > > Regarding the convergence of $u_t$, I am specifically referring to the authors' statements in lines 198-201:  "Konda & Tsitsiklis (1999) further assume that for any $c \in\mathbb{R}$, $w\in\mathbb{R}^d$, it holds that $\Phi w\neq ce$. Under this assumption, Tsitsiklis & Roy (1999) prove that $\Phi^{\top}D(P_{\pi} − I)\Phi$ is negative definite (Wu et al. (2020) assume this negative definiteness directly) and the iterates {wt} converges almost sure".
> > >
> > > An alternative approach would be to prove that $\Phi D(P_{\pi}-I)\Phi^{\top}$ is Hurwitz, for $\Phi$ describing the algorithm for $u$. Then the desired results are obtained using the ODE method.
> > >
> > > If the authors remain unconvinced, note that if Theorem 2 holds, then $u_t$ converges almost surely to a deterministic limit (which is the projection of $v_{\pi}$ onto the space with zero sum). Consequently, it is evident that proving almost sure convergence of a process to a deterministic limit is simpler than proving convergence that is sample-path-dependent.
> > >
> > > Although I have not performed a complete analysis, I believe that the perspective I am suggesting may allow for relaxed assumptions, particularly Assumption 4.4 on the learning rate and Assumption 4.5 on noise, which seem overly restrictive.
> > >
> > > Regarding the last paragraph in the authors' recent response, I agree with the form of the fixed point $v(\omega) = v_{\pi} + c(\omega)$. My point is that, with the perspective I propose, we have $c(\omega) = c_{\pi} + \lim_{t\to\infty} c_t(\omega)$, which is more precise since $c_t(\omega)$ can be computed during the algorithm.

---

> > > > ### Author Response · Authors · 2024-11-13
> > > >
> > > > We thanks the reviewer again for clarifying how to use Tsitsiklis & Roy (1999). The point we want to make is that, the ${u_t}$ updates rule the reviewer suggests doesn't seem to be related to the matrix $\Phi^\top D (P_\pi - I)\Phi$. Instead, according to our analysis in the previous reply, it is only related to the $A$ matrix (defined in the previous reply). But this $A$ matrix is not Hurwitz and does not satisfy the assumptions in Tsitsiklis & Roy (1999).
> > > >
> > > > The $u_t$ update is essentially the standard tabular average reward TD update plus some shift. So as described in our previous analysis, it's expected updates would be the summation between the expected updates of the standard TD, $D(P - I)u_t$, and the shift, $e^\top D(P - I) u_t e / |S|$. We cannot see how the $\Phi$ matrix would appear here.

---

> ### Comment · Reviewer_PPQC · 2024-11-13
>
> I disagree with the authors and feel they could have taken more time to attempt solving the issue themselves before asking for further clarification.
>
> The matrix of interest is not the one given in the authors' latest response. It is $A = \Phi^{\top}D(P - I)\Phi$, as I have already indicated. The remaining task is to determine the precise form of $\Phi$, which is not difficult but requires some effort. This is why I preferred to leave this task to the authors. Starting from the algorithm I described, we arrive at a slightly modified $\Phi$ compared to my initial suggestion. Specifically, we find that $\Phi^{\top} = (I_{|S|-1}, -e)\in\mathbb{R}^{(|S|-1)\times|S|}$.
>
> Using the same approach as Tsitsiklis & Roy (1999), we can then demonstrate that $A$ is negative definite. This, together with the ODE method, implies almost sure convergence.
>
> Ultimately, this proof is much more concise and straightforward than the one proposed by the authors, while requiring much weaker assumptions. If the authors still do not understand, I suggest they spend a few days working through the calculations themselves rather than asking me directly, as I am not their supervisor.

---

> > ### Author Response · Authors · 2024-11-14
> >
> > We appreciate the time and effort that the reviewer put in handling this submission and feel deeply privileged to have such a responsive reviewer. But we have to point out that the $A$ matrix constructed in the reviewer latest comment might be incorrect. To see this, we recall that to use the results in Tsitsiklis & Roy (1999), we have to compute the expected updates of $u_t$, i.e., to find the $A$ matrix such that
> > $$u_{t+1} = u_t + \alpha_{t+1} A u_t.$$
> > Here again we assume all rewards are 0 to simplify notations. It is our understanding that $u_t \in \mathbb{R}^{|S|}$. So for the update to make sense, the $A$ matrix has to be $|S|$ by $|S|$. But the $A$ matrix constructed by the reviewer is $|S| - 1$ by $|S| - 1$. So from a sanity check of the dimension, we believe the $A$ matrix constructed by the reviewer is wrong, unless we misunderstood the proposed algorithm by the reviewer.
> >
> > On the other hand, the $A$ matrix we constructed is $D(P - I)$. To convince the reviewer and to demonstrate that we have put due diligence into this, we attach a python script below that numerically verifies the correctness of our $A$. This script is self-contained and only needs numpy to run. A sample output is
> > ```
> > [[ 9.57424790e-05]
> >  [ 3.01789201e-04]
> >  [-3.97337865e-04]]
> > ```
> >
> > ```
> > import numpy as np
> >
> > def prob_measure(n, temp=1.0):
> >     logits = np.random.rand(n) / temp
> >     exp_logits = np.exp(logits)
> >     prob = exp_logits / np.sum(exp_logits)
> >     return prob
> >
> > def stochastic_matrix(n, temp=0.5):
> >     m = np.array([prob_measure(n, temp) for _ in range(n)])
> >     return m
> >
> > # Generate a stochastic matrix and compute its stationary distribution
> > # Each row of d is the stationary distribution
> > def P_and_d(n, temp, eps):
> >     P = stochastic_matrix(n, temp=temp)
> >     d = P
> >     while True:
> >         next_d = d @ P
> >         if np.mean(np.abs(d - next_d)) < eps:
> >             break
> >         d = next_d
> >     return P, d
> >
> > def test_A():
> >     n = 3
> >     all_states = np.arange(n)
> >     P, d = P_and_d(n, temp=1, eps=1e-5)
> >     d = d[0]
> >
> >     # randomly generate u_t
> >     u = np.random.rand(n, 1)
> >
> >     simulated_delta_u = np.zeros_like(u)
> >     trials = 100000
> >     for i in np.arange(trials):
> >         s = np.random.choice(all_states, p=d)
> >         next_s = np.random.choice(all_states, p=P[s, :])
> >         td_error = u[next_s, 0] - u[s, 0]
> >         simulated_delta_u += -1.0 / n * td_error
> >         simulated_delta_u[s, 0] += td_error
> >     simulated_delta_u = simulated_delta_u / trials
> >     true_delta_u = np.diag(d) @ (P - np.eye(n)) @ u
> >     print(simulated_delta_u - true_delta_u)
> >
> > if __name__ == '__main__':
> >     test_A()
> > ```

---

> ### Comment · Reviewer_PPQC · 2024-11-14
>
> Once again, I strongly advise the authors to take a few days to perform the calculations themselves, as it is increasingly evident that they do not fully grasp the main related works relevant to their submission.
>
> The changes I propose are not new nor original to me. They are a straightforward application of the work by Tsitsiklis & Roy (1999), with $\Phi$ chosen appropriately so that we can construct an alternative algorithm with two key properties: 1) it is equivalent to the algorithm proposed in this submission, meaning that one can derive the iterations of one from the other (with the need to track a few additional low-dimensional values, specifically the constant $c_t$ mentioned in my previous comments); 2) it yields a matrix $A=\Phi^{\top}D(P-I)\Phi$ that is negative definite.
>
> The idea is simple: the matrix $D(P-I)$ is not negative definite solely because of $e$, as we have $e^{\top}D(P-I)=0$. However, one can easily verify that this is the only problematic direction. In other words, $x^{\top}D(P-I)x < 0$ for any $x \in E - {0}$, where $E =${ $x \mid \sum_i x_i = 0 $}. Since $E$ is a $(|S|-1)$-dimensional vector space, the linear parametrization induced by the desired $\Phi$ should also be $(|S|-1)$-dimensional, making $\Phi$ of dimension $|S| \times (|S| - 1)$. Consequently, $A$ is clearly $(|S| - 1) \times (|S| - 1)$ dimensional, which makes sense, as the goal is to eliminate the single ill-conditioned direction.
>
> After reflecting more on the problem due to the authors’ inquiries, my opinion of their paper has shifted. In short, a straightforward application of existing literature yields the same results as those proved in the paper, but with significantly milder assumptions and much simpler proofs. I therefore believe the contributions of this paper are quite limited and indicate that the authors may lack a strong understanding of the primary literature in the field of their submission. It might be worth reconsidering submission in this domain.
>
> I will not provide further responses, as I have already dedicated considerable time to this review and provided ample guidance in my comments. For a competent researcher, these hints should be sufficient to derive a complete proof, provided they invest the necessary time and effort to work through it carefully.

---

> > ### Author Response · Authors · 2024-11-14
> >
> > We appreciate the reviewer's engagement and respect and understand the reviewer's decision of not providing further responses. This has been a long conversation and we fully agree with the reviewer that if some $\Phi$ satisfies the two properties that the reviewer raised then our results would be trivial.
> >
> > Yet, we also believe we have reached another agreement that the $u_t$ and $\Phi$ proposed by the reviewer in its current form do not satisfy those two properties. (This is just our reading of the reviewer's response. We quote "the linear parametrization induced by the desired $\Phi$ should also be $(|S| - 1)$-dimensional". But our understanding is that the proposed $u_t$ in its current form is $|S|$ dimensional. So we draw the conclusion that the current $u_t$ does not satisfy the proposed property.)
> >
> > As requested by the reviewer, we will spend extra days to investigate whether such a $\Phi$ exists or not.

---

> ### Comment · Reviewer_PPQC · 2024-11-14
>
> I believe the authors have misunderstood that $u_t$ is not itself the parameterization but rather the parametrized projected value function. Therefore, they need to introduce a separate parameterization, say $\theta \in \mathbb{R}^{|S|-1}$, such that $u = \Phi \theta$ resides in $E$, the $(|S|-1)$-dimensional subspace of $\mathbb{R}^{|S|}$ that I defined in my previous responses.

---

> > ### Author Response · Authors · 2024-11-14
> >
> > This is also what we believe. Let $A = \Phi^\top D(P - I)\Phi$, we then have
> > $$\theta_{t+1} = \theta_t + \alpha_t A \theta_t.$$
> > From Tsitsiklis & Roy (1999), it will be straightforward to conclude that $\theta_t$ will converge to 0.
> > Let $u = \Phi \theta$ and multiply $\Phi$ on both sides of the above equality. Then we get
> > $$u_{t+1} = u_t + \alpha_t \Phi \Phi^\top D(P - I) u_t.$$
> > Now the rest is to compare it with
> > $$v_{t+1} = v_t + \alpha_t D(P - I) v_t.$$
> >
> > We will definitely spend extra time comparing those two iterates but my initial assessment is that it's not very optimistic. Even comparing the deterministic expected updates is not that easy, not to say at the end we will need to compare the stochastic iterates.
> >
> > That being said, we do thank the reviewer for bringing up this $\theta_t$ algorithm which we are not aware of before. We acknowledge that this $\theta_t$ algorithm would be a valid algorithm to solve the policy evaluation problem in average reward MDPs. It's just that it might take quite some effort to establish the connection between this $\theta_t$ algorithm and the normal average reward TD. But we will definitely look into this possibility.

---

> > > ### Author Response · Authors · 2024-11-26
> > > **Analysis of the reviewer's proposal**
> > >
> > > As requested by the reviewer, we spent extra days analyzing the reviewer's proposal. Our conclusion is that the reviewer's method only works if we consider expected updates and the reward is always 0. It does not work if we consider the stochastic updates with general reward.
> > >
> > > In particular, for the special case, we agree with the reviewer's construction that $\phi^\top = (I, -ke)$ but this $k$ is not 1. We found that we need to use $k = \frac{-2 + \sqrt{4|S|}}{2(|S| - 1)}$. But for the general case, we do not think it will work.
> > >
> > > To demonstrate that we have done our due diligence, we revised the paper and would ask the reviewer to check our derivation (pages 31 - 35, Appendix D, at the very end of the submission), where we detailed our understanding of the reviewer's approach and the challenges that make it not work in general stochastic cases. We hope this could convince the reviewer that the result presented in this submission cannot be trivially derived from existing works. That being said, we would appreciate more hints on the stochastic cases if the reviewer still believes  that their proposed method would apply.

---

> ### Author Response · Authors · 2024-11-27
>
> We have revised the submission to provide another perspective to demonstrate that the reviewer's approach will not work. Let's assume for now it was somehow proved that $v_t = u_t + c_t e$ for the general stochastic updates with general reward (recall that this is actually not proved). We know that $\theta_t$ converges a.s., so $u_t$ converges a.s. to a fixed point. But our computation suggests that $c_t$ has a form of $c_t = \sum_{i=0}^{t-1} \alpha_i z^\top u_i$ for some fixed vector $z$ (see Appendix D.1 for the exact form of $z$). However, the convergence of $u_i$ does not imply the convergence of $c_t$. To proceed, we at least need the almost sure convergence rate of $u_t$ (this is far from enough). But to our knowledge, there is no almost sure convergence rate of average reward linear TD. If we cannot prove $c_t$ converge to a sample path dependent fixed scalar, we cannot prove that $v_t$ converge to a sample path dependent fixed point following the reviewer's approach. (We do note that our method does prove that $v_t$ converge to a sample path dependent fixed point).

---

> ### Comment · Reviewer_PPQC · 2024-12-03
>
> I disagree with the authors. The formulation of the update of $u$ straightforwardly implies the existence of $\Phi$. Still, it is true that none of my two $\Phi$ I proposed in my previous answers were working. Since the authors are still not convinced and other reviewers start doubting, here is a full proof relying essentially on basic linear algebra.
>
> Let us first construct $\Phi$.
>
> Let $J\in\mathbb{R}^{N\times N}$ be the matrix with only ones, i.e., $J_{i,j}=1$  for $1\leq i,j\leq N$. Observe that the matrix $I_N-J / N$ is symmetric and has two eigenvalues: $1$ with multiplicity $N-1$ and $0$ with multiplicity $0$. Therefore there exists $Q\in\mathbb{R}^{N\times N}$ an orthogonal matrix (i.e., $Q^{\top}Q=QQ^{\top}=I_N$) such that $I_N-J/N=Q\hat{D}Q^{\top}$ where $\hat{D}=$diag$(1,1,\dots,1,0)$, i.e., $\hat{D}$ is diagonal with $N-1$ ones and one zero.
> Define $\Phi\in\mathbb{R}^{N\times(N-1)}$ such that $\Phi_{i,j}=Q_{i,j}$ for $1\leq i\leq N$ and $1\leq j\leq N-1$ (i.e., $\Phi$ is $Q$ without its last column). Observe that:
> $$
> \Phi\Phi^{\top}=(I_N-J/N) \text{ and } \Phi^{\top}\Phi=I_{N-1}.
> $$
> Moreover, it is straightforward to check that Im$(\Phi)=E:=$Vect$(e)^{\top}$, so $\Phi$ is a one-to-one correspondence from $\mathbb{R}^{N-1}$ to $E$.
> Take $v_0\in\mathbb{R}^N$ any initial value for the iterative method proposed by the authors. Define $c_0=\frac1N\sum_{i=1}^Nv_i$ so that $v_0-c_0e\in E$, therefore there exists $\theta_0\in\mathbb{R}^{N-1}$ such that $\Phi\theta_0=v_0-c_0e$.
>
> Now, for $t\geq0$, define $\theta_t$ and $c_t$ by induction as
> $$
> \theta_{t+1}=\theta_t+\alpha_{t+1}\delta_{t+1}\phi(S_t)
> \text{ and }
> c_{t+1}= c_t+\alpha_{t+1}\delta_{t+1}/N
> \text{ where }
> \delta_{t+1}=R_{t+1}-J_t+\phi(S_{t+1})^{\top}\theta_t-\phi(S_t)^{\top}\theta_t.
> $$
> Define $u_t=\Phi \theta_t$ as the approximated value function. A straightforward induction implies that $v_t=u_t+c_te$.
>
> Now it only remains to prove that $A=\Phi^{\top}D(P_{\pi}-I_N)\Phi$ is Hurwitz. It is sufficient to observe that $\Phi w\in E$ is orthogonal to $e$ for any $w\in\mathbb{R}^{N-1}$ and to cite the sentences of the authors line 197-201 : "Nevertheless, to proceed with the theoretical analysis, besides the standard assumption that $\Phi$ has linearly independent columns,  Tsitsiklis & Roy (1999); Konda & Tsitsiklis (1999) further assume that for any $c\in\mathbb{R}$, $w\in\mathbb{R}^d$, it holds $\Phi w\neq ce$. Under this assumption, Tsitsiklis & Roy (1999) prove that $\Phi^{\top}D(P_{\pi}-I_N)\Phi$  is negative definite (Wu et al. (2020) assume this negative definiteness directly) and the iterate [{$\theta_t$}]  converges almost surely."

---

> > ### Comment · Reviewer_PPQC · 2024-12-03
> >
> > Therefore, if the paragraph in lines 197–201 is correct, the proof provided in my previous answer demonstrates that the results in this submission have already been established in the literature in a much stronger form. This is because most of the assumptions on page 6 are unnecessary, and the proof is both simpler and clearer.
> >
> > That being said, I maintain my score. Nonetheless, I apologize to the authors for being so harsh: I acknowledge their work and effort, but I sincerely doubt that this work could or should be published, even with major revisions, in any well-ranked conferences or journals.

---

> > > ### Author Response · Authors · 2024-12-03
> > >
> > > We thank the reviewer again for the detailed response. We agree and are glad to see that this equivalence can be established. We feel deeply privileged to have interacted with such a responsible and smart reviewer. We believe this long conversation would be very useful for the community, after all this equivalence is not documented before anywhere else.
> > >
> > > That being said, we do believe one more step is still missing. In particular, we still need to prove that $c_t(\omega)$ converges to some $c(\omega)$. Otherwise, from the reviewer's derivation, we can at most say $v_t$ converge to a (possibly unbounded) set of fixed points almost surely. But we proved $v_t(\omega)$ converges to some $v(\omega)$. So it would be much appreciated if the reviewer could point out to some results that can be used to prove the convergence of $c_t$.
> > >
> > > By telescoping $c_t$, we can get $$c_t = c_0 + \sum_{i=0}^{t-1} \alpha_{i+1} \delta_{i+1} = c_0 + \sum_{i=0}^{t-1} \alpha_{i+1}(R_{i+1} - J_\pi + \phi_{i+1}^\top \theta_* - \phi_{i}^\top \theta_*) + \sum_{i=0}^{t-1} \alpha_{i+1} (J_\pi - J_t) + \sum_{i=0}^{t-1} \alpha_{i+1}(\phi_{i+1}^\top - \phi_i^\top) (\theta_t - \theta_*).$$
> > > Here we use $\phi_i \doteq \phi(S_i)$ and use $\theta_*$ to denote the limit of $\theta_t$.
> > > The first summation converges to 0 due to ergodic theorem. The second also converges because we have the almost sure convergence rate for $J_t \to J_\pi$ (This rate will restrict the choice of $\alpha_i$). But for the third summation, it is not clear to us whether it converges or not, essentially because we do not have the almost sure convergence rate of $\theta_t$. We can alternatively write $c_t$ as
> > > $$c_{t+1} = c_t + \alpha_{t+1} (f(c_t) + \delta_t)$$ with $f(c_t) \equiv 0$ to match the form of standard stochastic approximation algorithms. But the problem is the noise $\delta_t$ does not diminish. If we want to use standard SA results, e.g., [1], we need to bound the total accumulated noise $\sum_i \alpha_i \delta_i$. Without the almost sure convergence rates of $\theta_t \to \theta_*$, it is not clear to us how this can be done. Alternatively, this problem can also be solved if we can somehow prove that $\sup_t c_t < \infty$ a.s.. We are wondering if the reviewer could provide more hints on this to conclude this long conversation.
> > >
> > > [1] Vivek S. Borkar. Stochastic approximation: a dynamical systems viewpoint. Vol. 9. Cambridge: Cambridge University Press, 2008.

---

### Official Review · Reviewer_P1wx · 2024-11-03

**Soundness:** 3
**Presentation:** 3
**Contribution:** 3
**Rating:** 6
**Confidence:** 3

**Summary:**

The paper studied the convergence temporal difference(TD) algorithm in average reward setting and tabular case. What differs the work from many existing literature is the focus on almost sure (a.s.) convergence. Discussions on limitations of existing approaches on a.s. is provided with clarity. A general a.s. convergence for stochastic system with Markovian noise and additive noise is established, as a result of which TD algorithm in average reward setting is established.

**Strengths:**

* A long standing a.s. convergence of TD in average reward setting is investigated and convergence result is confirmed.

* A general a.s. general a.s. convergence for stochastic system with Markovian noise and additive noise is established, which might be of an independent interest.

* Discussions on existing approaches are helpful and provide clarity.

**Weaknesses:**

* Although an a.s. convergence is characterized, it's not clear as to how the connection of sample path and the resulting fixed point. It would be nice to have further clarification.

* In the sentence of Line 149-151, the authors argued that existing analysis failed to move beyond convergence to a bounded invariant set. However, the main result in Theorem 2 falls into the same category, as it converges to $\mathcal{V}_{*}$, which is an unbounded set.

**Questions:**

1. The value function defined in Line 65 appears slight different from Page 250 value function definition in [1], are they equivalent? If so how to see this?

[1] Sutton, Richard S. "Reinforcement learning: An introduction." A Bradford Book (2018).

2. In equation (5), why not just cancel out v(t) with the last term in h(v(t))? Is there some consideration for not doing so?

3.  In the sentence of Line 149-151, the authors argued that existing analysis failed to move beyond convergence to a bounded invariant set. Are there any particular references the authors are referring to? If so, please provide them.

4. In Line 168, it mentioned that "it cannot be used once function approximation is introduced". Can you elaborate more on this?

5. Maybe a question concerns future effort, where are the potentially challenges lie in analysis in order to achieve a convergence with rate rather asymptotic convergence?

---

> ### Author Response · Authors · 2024-11-27
> **Response to Reviewer P1wx**
>
> > Weaknesses: Although an a.s. convergence is characterized, it's not clear as to how the connection of sample path and the resulting fixed point. It would be nice to have further clarification. In the sentence of Line 149-151, the authors argued that existing analysis failed to move beyond convergence to a bounded invariant set. However, the main result in Theorem 2 falls into the same category, as it converges to V^*  which is an unbounded set.
>
> We would like to clarify the distinction between converging to a bounded invariant set and converging to a fixed point within that set. If only convergence to an invariant set is established, the value function estimate may oscillate indefinitely among different values within that set. In contrast, our analysis proves that the iterates converge to a specific fixed point within the invariant set.
>
> When we refer to this as a "sample-path dependent fixed-point," we mean that the specific point of convergence within the invariant set depends on the initialization of the value function and the realization of the random Markovian noise process $Y_t$ and cannot be determined a-priori. While the resulting fixed-point may depend on the trajectory, the key contribution of our analysis is proving that the iterates settle to some stable, specific point within the invariant set, rather than merely remaining bounded.
>
> > Questions:
> The value function defined in Line 65 appears slight different from Page 250 value function definition in [1], are they equivalent? If so how to see this?
>
> The definitions of the value function are equivalent. If you look at equation 10.9 we can see how they define the returns $G_t$ in the average reward setting. This is equivalent to our definition on line 65 (except we follow the convention where the average reward is denoted as $\bar{J_\pi}$ instead of $\bar{r}$). Then we can use the standard definition of the value function which is: v(s) = $E_{\pi}[G_t|S_t = s]$ which Sutton gives in the paragraph that follows equation 10.9.
>
> >In equation (5), why not just cancel out v(t) with the last term in h(v(t))? Is there some consideration for not doing so?
>
> The reason is purely for notation purposes. If you look at the form of equation 3, we want to write it to match our SKM iteration form in Equation (SKM with Markovian and Additive Noise). Therefore we want to keep the extra $v_t$ outside of $H$, which means it remains outside of $h$, which is the expectation of $H$. Thats why in equation 5 it remains outside and we don't cancel it.
>
> >In the sentence of Line 149-151, the authors argued that existing analysis failed to move beyond convergence to a bounded invariant set. Are there any particular references the authors are referring to? If so, please provide them.
>
> Thank you for pointing this out. To clarify, when we state that "existing ODE-based convergence analysis fails to move beyond convergence to a bounded invariant set," we are not implying that there are specific works using the ODE method to prove convergence to a bounded invariant set for average reward TD. Rather, we mean that existing **ODE-based analysis methods** are limited in that they  cannot provide results beyond convergence to a bounded invariant set in this context. We have revised the paper to reflect this more precise wording.
>
> >In Line 168, it mentioned that "it cannot be used once function approximation is introduced". Can you elaborate more on this?
>
> Count-based learning rates rely on state visitation counts to dynamically adjust the step size during updates. However, with function approximation, it's usually assumed that we have access to only features, not the state itself. Since the feature is usually not a one-to-one mapping, there is no way to count the state visits.

---

> ### Author Response · Authors · 2024-11-27
> **Response to reviewer P1wx (part 2)**
>
> >Maybe a question concerns future effort, where are the potentially challenges lie in analysis in order to achieve a convergence with rate rather asymptotic convergence?
>
> This is an excellent question. Using our results on stochastic Krasnoselskii-Mann iterations with Markovian noise, we were actually able to establish a convergence rate for the **expected residuals**, $E[\||T v_t - v_t\||_\infty]$, of our iterates where $T$ refers to the differential bellman operator. In the discounted setting, this convergence rate could directly translate to a convergence rate to the optimal value function due to the contraction mapping property of the Bellman operator. However, in the average reward setting, $T$ is only non-expansive in the infinity norm, rather than being a contraction. This prevents us from converting the residual convergence rate into a convergence rate to the optimal value function. We did not include the expected residual convergence result in the paper because it is not a stronger theoretical result than the $L_2$ convergence rate provided in Zhang et al. (2021). Furthermore, our almost sure convergence result demonstrates convergence to a sample-path dependent fixed point, meaning that the iterates converge to a potentially random point in $V^*$. Since the specific point of convergence depends on the random realization of the Markovian noise and the initialization, we could not derive a universal convergence rate that holds across all sample paths.

---

### Official Review · Reviewer_cv9m · 2024-11-04

**Soundness:** 3
**Presentation:** 3
**Contribution:** 1
**Rating:** 3
**Confidence:** 5

**Summary:**

The authors present an almost sure convergence analysis for finite state-action spaces, infinite horizon average reward in the tabular setting (that is without linear function approximation for the relative value function). There are many prior works in this regime, but the ones which are the most relevant are: (i) Tsitsiklis and Van Roy 1999 and (ii) Zhang et al 2021. The problem formulation addresses data sets obtained through Markovian sampling, with additive noises.

**Strengths:**

It is theoretically interesting to examine the almost sure convergence of fixed policy evaluation in tabular settings without relying on value function projection to achieve the contraction needed for analysis.

**Weaknesses:**

The amount of literature in this realm is quite significant. My main issue is the limited scope of this work. For instance, the following have been very well studied:
(i) Tsitsiklis and Van Roy 1999 consider policy evaluation using linear value function approximation, where the feature vectors do not span the constant vector (a mild assumption which doesn't hinder the applicability of their approach in many problems). They provided an asymptotic convergence analysis for this TD learning algorithm. The authors argue that their assumption does not hold when considering tabular cases and that might be true, but for most applications of interest, the state and action spaces are large enough to necessitate the use of function approximations for value function estimation to ensure practicality. Hence, the importance of this work is not well motivated.
(ii) Zhang et al 2021 considered the Tsitsiklis and Van Roy 1999 approach and relaxed the assumption by including a projection step, where the value function vectors are projected onto a subspace where an unique representation for them exists (generally the average reward value functions aren't unique vectors and are unique only upto an additive constant, and this projection eliminates this non uniqueness). They later characterize convergence in L2 space instead of asymptotic convergence and provide finite time bounds in terms of expectations of quantity of interest. The authors argue L2 convergence does not imply almost sure convergence which is true and also claim that the iterates converge to a set instead of a unique point. But if every point in this set is a solution, there is no need to obtain a unique solution since the original value function is not unique anyway.

Given all these limitations, I feel the work lacks sufficient contribution.

**Questions:**

Why is the asymptotic analysis for only the tabular case important? Prior results exist for applications with linear function approximations which capture almost all applications of interest. And prior literature also provides finite time bounds (although in the L2 norm sense) which are of more significance in terms of determining sample complexities, etc. The motivation for this work is not compelling in its current form.

---

> ### Author Response · Authors · 2024-11-27
> **Response to Reviewer cv9m**
>
> > The amount of literature in this realm is quite significant. My main issue is the limited scope of this work. For instance, the following have been very well studied: \
> > (i) Tsitsiklis and Van Roy 1999 consider policy evaluation using linear value function approximation, where the feature vectors do not span the constant vector ...The authors argue that their assumption does not hold when considering tabular cases and that might be true, but for most applications of interest, the state and action spaces are large enough to necessitate the use of function approximations for value function estimation to ensure practicality. Hence, the importance of this work is not well motivated.
>
> While we agree that function approximation is critical in many practical applications, the tabular setting remains a foundational component of RL. By rigorously addressing this setting, our work not only resolves an open problem but also contributes to a deeper understanding of TD learning, corrects misconceptions in the field, and establishes a solid theoretical basis for future advancements.
>
> 1. **Clarifying Misconceptions:** As demonstrated by our discussion with Reviewer PPQC, there is a prevalent misunderstanding in the research community that convergence results for average reward TD with linear function approximation automatically imply convergence in the tabular case. This is incorrect. A similar error also appears in a recent work *Natural Policy Gradient for Average Reward Non-Stationary RL,* (submission ID 12405 for ICLR 2025) which also mistakenly made this conclusion, as evidenced in their discussions with reviewers and subsequent revisions. The authors of that work have since acknowledged that almost sure convergence for linear function approximation does not imply convergence in the tabular setting. By explicitly addressing and resolving this gap, our work removes confusion and contributes a rigorous foundation for policy evaluation in the tabular setting.
>
> 2. **Distinct Theoretical Contributions:** From a theoretical standpoint, convergence for linear function approximation does not subsume the tabular case. The tabular setting presents unique challenges that require distinct techniques and insights. For instance, analogous challenges exist in other fields, such as topology: while Poincaré’s conjecture is straightforward in higher dimensions, its proof for $d = 3$ was exceptionally challenging and significant. Similarly, resolving convergence in the tabular case improves our theoretical understanding of temporal difference (TD) learning.
>
> 3. **Broader Significance of Our Mathematical Results:** Beyond addressing convergence for average-reward TD in the tabular setting, our work extends the Stochastic Krasnoselskii-Mann (SKM) iterations to include Markovian noise, providing a general mathematical result that has broader applications for proving the convergence of other RL algorithms.
>
> > (ii) Zhang et al 2021 considered the Tsitsiklis and Van Roy 1999 approach and relaxed the assumption ... They later characterize convergence in L2 space instead of asymptotic convergence and provide finite time bounds in terms of expectations of quantity of interest. The authors argue L2 convergence does not imply almost sure convergence which is true and also claim that the iterates converge to a set instead of a unique point. But if every point in this set is a solution, there is no need to obtain a unique solution since the original value function is not unique anyway.
>
> We acknowledge the contributions of Zhang et al. (2021) in extending the Tsitsiklis and Van Roy (1999) framework by introducing a projection step. However, we maintain that L2 convergence does not imply almost sure convergence, and this distinction is crucial both theoretically and practically. To illustrate this, define a sequence of independent random variables ${z_n}$ such that $Pr(z_n = n^\nu) = \frac{1}{n}$ for $\nu < \frac{1}{2}$, and $Pr(z_n = 0) = 1- \frac{1}{n}$. Then we have $E[z_n^2] = n^{2\nu -1}$, so the sequence converges in $L^2$ for $\nu < \frac{1}{2}$. However, we have $\sum_n Pr(z_n = n^\nu) = \infty$. By the second Borel-Cantelli lemma, $z_n$ diverges to $\infty$ almost surely. This example demonstrates that $L^2$ convergence alone does not guarantee the behavior of individual realizations, a critical issue in contexts like continual learning, where a single trial dictates performance. Practically, almost sure convergence is essential for understanding the long-term behavior of learning algorithms, especially in real-world applications where only a handful of trajectories are observed.
>
> While it is true that every point in the set to which the iterates converge is a solution, the lack of of a fixed point is also important. Without almost sure convergence to a specific point within the set, the the value function representation remains unstable which can complicate practical implementations.

---

### Meta-Review · Area_Chair_xZ3k · 2024-12-07

**Metareview:**

The paper studies the convergence behavior of the temporal difference (TD) algorithm for the average reward MDP (AMDP), under the tabular setting where the state and action spaces are finite. By extending the recent convergence results in the Stochastic Krasnoselskii-Mann iterations to the settings with Markovian and additive noise, the authors show an almost sure convergence of the tabular TD method to a sample-path-dependent fixed point. The paper certainly addresses some gap in the a.s. convergence of TD algorithm for ADMP, yet the contribution does not seem enough as most of the results are based on the well-established techniques of SKM iteration and Markovian noise. The discussion with existing literature is also not enough, in which stronger results have already been established.  Therefore, we decide to reject this paper.

**Additional Comments On Reviewer Discussion:**

Except for some minor changes, the main theme of this rebuttal is about whether the proposed results are implied by the existing literature. The result is not completely dominated by existing works, but the remaining gap is not as large as what the authors claimed, and the analysis can be much simpler following Reviewer  PPQC's suggestion.

---

### Decision · Program_Chairs · 2025-01-22

Reject